# Potential contribution to secondary aerosols from benzothiazoles in the atmospheric aqueous phase based on oxidation and oligomerization mechanisms

Qun Zhang<sup>1,2</sup>, Wei Zhou<sup>1</sup>, Shanshan Tang<sup>4</sup>, Kai Huang<sup>1</sup>, Jie Fu<sup>1</sup>, Zechen Yu<sup>1</sup>, Yunhe Teng<sup>1</sup>, Shuyi Shen<sup>1</sup>, Yang Mei<sup>1</sup>, Xuezhi Yang<sup>1</sup>, Jianjie Fu<sup>1,2,3,\*</sup>, Guibin Jiang<sup>1,2,3</sup>

<sup>1</sup>School of Environment, Hangzhou Institute for Advanced Study, University of Chinese Academy of Sciences, Hangzhou 310024, China

<sup>2</sup>State Key Laboratory of Environmental Chemistry and Ecotoxicology, Research Center for Eco-Environmental Sciences, Chinese Academy of Sciences, Beijing 100085, China

<sup>3</sup>College of Resources and Environment, University of Chinese Academy of Sciences, Beijing 100049, China

Correspondence to: Jianjie Fu (jjfu@rcees.ac.cn)

Abstract. Benzothiazoles (BTs), widely used as vulcanization accelerators in the rubber industry, have frequently been identified in the atmosphere, especially in areas with heavy traffic. BTs can undergo gas-phase oxidation in the atmosphere. which contributes to secondary aerosol mass. However, given their certain water-solubility atmospheric fate of BTs associated with aqueous-phase transformations is unclear. In this study, the reactions of benzothiazole (BT), 2methylbenzothiazole (MBT), and 2-chlorobenzothiazole (CBT), with hydroxyl radicals (OH) were investigated. The rate constants of BT, MBT, and CBT reacted with OH radicals were determined to be  $(8.0 \pm 1.8)$ ,  $(7.6 \pm 1.7)$ , and  $(7.6 \pm 1.9) \times$  $10^9 \text{ M}^{-1} \text{ s}^{-1}$  at initial pH 2, and  $(9.7 \pm 2.7)$ ,  $(9.8 \pm 2.7)$ , and  $(9.4 \pm 2.7) \times 10^9 \text{ M}^{-1} \text{ s}^{-1}$  at initial pH 10, respectively. Lifetimes ranging from several minutes to several hours were estimated under mean OH concentrations in various atmospheric aqueous phases, which are significantly shorter than those estimated under mean OH concentrations in the gas phase. The nanoparticle tracing analysis (NTA) directly shows the formation of nanoparticles from the aqueous phase photooxidation of the selected BTs. Data analysis of liquid chromatography Orbitrap mass spectrometry (LC-Orbitrap MS) identifies many multifunctional oligomers. Changes in optical property support the formation of oligomers and suggest that the products have the potential to contribute to the atmospheric brown carbon. In addition, higher yields of sulfate are formed after the reactions. It is highlighted that the aqueous-phase oxidation of BTs can contribute to the secondary aerosol mass in the ambient atmosphere, particularly in polluted regions where BTs concentrations are comparable to those of benzenes, potentially altering the chemical composition and optical properties of atmospheric particles.

## 1 Introduction

Benzothiazoles (BTs) are a class of aromatic heterocyclic organic compounds with a thiazole ring fused to a benzene ring.

BTs are categorized as high-production-volume chemicals, with a reasonable annual global production estimate of hundreds

<sup>&</sup>lt;sup>4</sup>Hangzhou International Innovation Institute, Beihang University, Hangzhou 311115, China

of thousands of tons (Liao et al., 2018). They are used in a variety of industrial and consumer products, mainly including vulcanization accelerators in rubber production, ultraviolet light stabilizers in textiles and plastics, and precursors in the production of pharmaceuticals (Avagyan et al., 2015; Reddy and Quinn, 1997). BTs can be also biogenically produced from plants and microorganisms(De Wever and Verachtert, 1997; Chhalodia et al., 2021; Stierle et al., 1991). The high production, extensive use, and decentralized presence of BTs have resulted in their widespread release into the environment by volatilization, leaching, and abrasion (Luongo et al., 2016; Parker-Jurd et al., 2021; Zhang et al., 2018a; Schneider et al., 2020; Tang et al., 2022a; Bayati et al., 2021; Armada et al., 2022). Moreover, BTs have been commonly detected in human blood, urine, adipose tissue, breast milk and even amniotic fluid (Liao et al., 2018; Garcia-Gomez et al., 2015; Li et al., 2022; Ferrario et al., 1985; Zhou et al., 2020). BTs have the potential to disrupt the endocrine system, and their exposure in the body may pose a certain risk to human health (Chen et al., 2020; Cao et al., 2023; Zhou et al., 2020; Li et al., 2022). The German Environmental Protection Agency has classified BTs as potentially persistent, mobile and toxic substances, making them highly concerning (Neumann and Schliebner, 2019).

35

Commonly used BTs have been detected in the atmosphere of cities worldwide, with ΣBTs concentrations ranging from tens of pg m<sup>-3</sup> to several μg m<sup>-3</sup> (Zhang et al., 2025; Johannessen et al., 2022; Chang et al., 2021; Wu et al., 2021; Nuñez et al., 2020; Dörter et al., 2020; Wei et al., 2025). The mean concentration of Σ<sub>4</sub>BTs in 18 major cities worldwide is 66 pg m<sup>-3</sup>, higher than the four types of flame retardants Σ<sub>9</sub>PBDEs (polybrominated diphenyl ethers), Σ<sub>11</sub>NFRs (novel flame retardants), HBCDD (hexabromocyclododecane), and TBBPA (tetrabromobisphenol A) with mean values of 42, 42, 15, and 17 pg m<sup>-3</sup>, respectively (Johannessen et al., 2022; Saini et al., 2020). The median concentration of Σ<sub>6</sub>BTs in PM<sub>2.5</sub> in typical cities of China ranges from 305 to 564 pg m<sup>-3</sup>, significantly higher than another widely concerned rubber additive Σ<sub>6</sub>PPDs (*p*-phenylenediamine antioxidants, with median concentrations of 27-103 pg m<sup>-3</sup>) (Liao et al., 2021; Zhang et al., 2022). Benzothiazole has the highest content amongst all BTs species in the atmosphere, whose concentration in the gas phase can reach 3.86 μg m<sup>-3</sup> in some region, larger than or equivalent to the common anthropogenic species, such as benzene, toluene, xylenes, benzaldehyde, etc (Dörter et al., 2020). The highest concentration of PM<sub>2.5</sub>-bound BTs can also reach to a level of several ng m<sup>-3</sup> (Liao et al., 2021; Zhang et al., 2020b; Dörter et al., 2020). In addition, BT is also present in urban rain, with a concentration as high as 70 ng L<sup>-1</sup> (Ferrey et al., 2018).

Owing to their extensive use and widespread in the atmosphere, the environmental fate of BTs associated with atmospheric transformations began to be brought into focus in recent years. The gas-phase oxidation of benzothiazole by OH radicals has been experimentally and theoretically investigated, in which the rate constant has been determined to be of  $(2.1 \pm 0.1) \times 10^{-12}$  cm<sup>3</sup> molecule<sup>-1</sup> s<sup>-1</sup>, and the mechanism for the reaction of OH with benzothiazole via the initial attack on different carbon sites has been proposed (Karimova et al., 2024). Simulation experiments of the gas-phase benzothiazole photooxidation with OH radicals indicated that benzothiazole could contribute to secondary organic aerosols after their oxidation into C3-8 organic compounds and could also lead to the production of sulfuric acid (Franklin et al., 2021). Aqueous phases are also significant reaction media in the ambient atmosphere, including clouds, fogs, or wet aerosol particles (Blando and Turpin, 2000; Herrmann et al., 2015; Ervens, 2015; McNeill, 2015). OH radicals are both highly reactive and nonselective in the

atmospheric aqueous phase compared to other atmospheric oxidants, which can significantly shorten the atmospheric lifetime of pollutants (Herrmann, 2003). Based on laboratory, modeling, and field studies conducted over the past decades, there has been increasing evidence that aqueous-phase processing affects the formation of both organic and inorganic components of secondary aerosols and contributes to the production of light-absorbing matter, which has the potential to influence climate (Lv et al., 2025; Mei et al., 2025; Li et al., 2023b; Zhang et al., 2018b; Ervens, 2015; Herrmann et al., 2015; Laskin et al., 2015; McNeill, 2015; Zhang et al., 2020a; Meagher et al., 1990). BTs are water-soluble, as the water solubility of benzothiazole is experimentally determined to be 4300 mg L<sup>-1</sup> at 25 °C (Wishart et al., 2022). It is higher than or comparable to many compounds emitted from biomass burning those have been proved to undergo aqueous-phase processes and contribute to secondary organic aerosols in the atmosphere, such as syringic acid (1548 mg L<sup>-1</sup> at 25 °C), α-terpincol (710 mg L<sup>-1</sup> at 25 °C), and vanillic acid (2791 mg L<sup>-1</sup> at 25 °C) (Li et al., 2021; USEPA, 2012; Li et al., 2020; Tang et al., 2020b). Therefore, understanding the photooxidation reactions of BTs in the aqueous phase is significant not only for the integrated determination of the atmospheric fate of BTs, but also for assessing their potential contributions to both organic and inorganic components of secondary aerosols due to their nature as sulfur-containing organic compounds. However, the detailed aqueous-phase oxidation mechanisms of BTs in the atmosphere remain largely unexplored, hindering a holistic understanding of their atmospheric chemistry and potential health risks.

The objective of this study is to investigate the aqueous-phase transformation of BTs by OH radicals in the atmosphere. Therefore, photooxidation reactions of OH radicals and frequently detected BTs in the atmosphere, such as benzothiazole (BT), 2-methylbenzothiazole (MBT), and 2-chlorobenzothiazole (CBT) (Fig. 1) (Liao et al., 2021; Nuñez et al., 2020; Dörter et al., 2020), were carried out at room temperature in an aqueous solution using a photoreactor. Kinetics of reactions of selected BTs and OH radicals are determined. Products from reactions of selected BTs and OH radicals were analyzed, based on which the reaction mechanisms were proposed.

Figure 1. Structures, formula, and molecular weights of benzothiazole (BT), 2-methylbenzothiazole (MBT), and 2-chlorobenzothiazole (CBT) (white balls represent H atoms).

### 2 Materials and methods

#### 2.1 Chemicals


Benzothiazole (BT, > 96.0%), 2-methylbenzothiazole (MBT, > 98.0%), 2-chlorobenzothiazole (CBT, > 98.0%), p-toluic acid (TA, > 98.0%), L-phenylalanine (LPA, > 98.0%), and sodium hydroxide (NaOH, 1 mol L<sup>-1</sup> in water) were purchased

from Tokyo Chemical Industry (Japan). Suberic acid (SA,  $\geq$  99%) was purchased from Adamas (Swiss). Perchloric acid (HClO<sub>4</sub>, 0.1 mol L<sup>-1</sup> in water) was purchased from Aladdin (China). Hydrogen peroxide (H<sub>2</sub>O<sub>2</sub>, 30% wt in water) was purchased from Sinopharm (China). All chemicals were used as received without further purification. Ultrapure water (18.2 M $\Omega$  cm at 25°C) was supplied by a Milli-Q water purification system. All solutions were freshly prepared for each experimental run.

# 2.2 Aqueous phase photoreactor

Aqueous photochemical experiments were conducted in a 250 mL quartz glass vessel with a path length of 7 cm, and 200 mL of solution was initially used for the reaction. The solution was continuously stirred with a glass magnetic stir bar and irradiated with a UVB lamp (peak emission 306 nm, 6W, G6T5E, Sankyo-Denki, Japan) whose wavelength falls into the typical actinic region (> 290 nm) of the solar spectrum for generation of OH radicals through H<sub>2</sub>O<sub>2</sub> photolysis to trigger oxidation reactions. All experiments were carried out at 298 ± 1 K.

### 2.3 Kinetic experiments




A relative rate technique is widely used for the investigation of aqueous-phase kinetics as it does not involve the concentration of OH radicals, and the reactant concentration does not need to be accurately known (Aljawhary et al., 2016). Therefore, the relative rate method was used to determine the second-order rate constants of the reaction of BT, MBT, and CBT (denoted as BTs) with OH radicals in this study. This method relies on the assumption that BTs and the reference compounds are removed solely by reacting with OH radicals. The rate constant was obtained by detecting the relative losses of BTs and the reference compounds with well-known rate constants of reactions with OH radicals. The rate constant was calculated by Eq. (1).

$$ln\frac{[BTs]_0}{[BTs]_t} = \frac{k_{BTs}}{k_{Ref}}ln\frac{[Ref]_0}{[Ref]_t}$$

$$\tag{1}$$

In Eq. (1),  $[BTs]_{\theta}$  and  $[BTs]_{t}$  are the concentrations of BTs at time zero and time t;  $[Ref]_{\theta}$  and  $[Ref]_{t}$  are the concentrations of reference compounds at time zero and time t; and  $k_{BTs}$  and  $k_{Ref}$  are second-order rate constants (M<sup>-1</sup> s<sup>-1</sup>) for reactions of OH radicals and compounds of interest and reference, respectively.

In the real atmosphere, the range of pH values in the aqueous phase is from highly acidic to slightly alkaline in both dilute cloud droplets and concentrated deliquescent particles (Herrmann et al., 2015). Therefore, initial pH 2 and 10 were chosen to simulate the aqueous phase in the atmosphere for kinetic studies. In experiments for rate constants determination, the reaction mixture was a 1-2 μM aqueous solution of a kind of BTs with total volume of 100 mL. The initial pH of the reaction solution was adjusted to 2 or 10 with HClO<sub>4</sub> or NaOH, which was measured by a pH meter (SevenCompact, Mettler Toledo). 1 mM of H<sub>2</sub>O<sub>2</sub> in the reaction mixture was photolyzed with UVB irradiation to generate OH radicals. Two reference compounds were used under each pH condition, of which rate constants are listed in Table 1 (Buxton et al., 1988). Kinetic

experiments were performed in triplicate at initial pH 2 and 10, with the concentration ratio of BTs to the reference compounds adjusted to ensure accurate determination of the rate constants for the reactions of BTs with OH radicals.

The uncertainty of the determined rate constant of BTs reacted with OH radicals was derived from the uncertainty of the linear fit of the plot of  $ln([BTs]_0/[BTs]_t)$  versus  $ln([Ref]_0/[Ref]_t)$  and the error of the rate constant of the reference compound (Kramp and Paulson, 1998). The uncertainty of the slope was twice the standard error (Tang et al., 2020b; Li et al., 2021). It was mainly attributed to the analytical instrument for determination of relative losses of BTs and reference compounds (Messaadia et al., 2013). Uncertainties of reference compounds were estimated at 10% according to a previous study (Buxton et al., 1988).

Table 1. Second-order rate constants for the aqueous-phase oxidation of reference compounds by OH radical (Buxton et al., 1988).

| Reference compounds | k for oxidation by OH (M <sup>-1</sup> s <sup>-1</sup> ) $\times$ 10 <sup>9</sup> |               |  |  |
|---------------------|-----------------------------------------------------------------------------------|---------------|--|--|
|                     | pH = 2                                                                            | pH = 10       |  |  |
| Suberic acid        | $4.8 \pm 0.4$                                                                     | -             |  |  |
| L-phenylalanine     | $5.7 \pm 0.5$                                                                     | $9.0 \pm 0.9$ |  |  |
| p-Toluic acid       | -                                                                                 | $8.0 \pm 0.8$ |  |  |

During a kinetic experiment, 1-mL aliquots were sampled from the reaction beaker at designated time intervals, in which a total of 10 mL solution was removed from the 100 mL solution with a negligible effect on the determination of kinetics. Relative losses of BTs and reference compounds were immediately detected with a liquid chromatograph (LC, Vanquish, Thermo Scientific, USA) coupled with a triple-quadrupole mass spectrometer (MS, TSQ Altis Plus, Thermo Scientific, USA) equipped with an electrospray ionization (ESI) source. The multiple reaction monitoring (MRM) mode was used in the MS operation, and the full description of the LC-MS analysis method has been provided in the Supplement (Sec. S1). Control experiments were conducted to demonstrate that the BTs degradation was only attributed to their oxidation by OH radicals. Experiments without UVB illumination were to ensure that H<sub>2</sub>O<sub>2</sub> would not degrade BTs, and photodegradation experiments without the addition of H<sub>2</sub>O<sub>2</sub> were to determine if BTs would undergo direct photolysis. Results of control experiments confirmed that the decay of BTs did not occur during the period of kinetic experiments (Fig. S1).

### 2.4 Experiments for products formation





In experiments for products formation, pH 2 and 10 were also chosen to simulate the aqueous phase in the atmosphere. The reaction solution consisted of a kind of BTs (~0.45 mM), H<sub>2</sub>O<sub>2</sub> (10 mM), and initial pH adjuster (HClO<sub>4</sub> or NaOH) with a total volume of 200 mL. During an experiment, 10-mL aliquots were sampled from the reaction beaker at time intervals of one hour for subsequent analysis. BTs concentrations were obtained with LC-MS whose method is described in the Supplement (Sec. S1). The experiments were performed for 5 h. The steady-state concentration of OH radicals can be estimated through dividing the BTs loss by the bimolecular rate constant of BTs with OH radicals (Eq. (2); George et al.,

2015). Considering the different parent compounds and the rate constants at pH 2 and 10, the steady-state concentration of OH radicals was estimated to be approximately 10<sup>-14</sup> M.

$$[OH] = \frac{k'}{k_{BTS + OH}} \tag{2}$$

# 2.5 Measurements of optical properties





Ultraviolet-visible (UV-vis) absorption spectra of the samples collected at time intervals of one hour were obtained by a UV-vis spectrophotometer (UV-2700i, Shimadzu, Japan) in the range of 200-800 nm with a step size of 1 nm. The sample for UV-vis spectrum measurement was contained in a quartz cuvette with an optical length of 10 mm. Ultrapure water was measured to obtain the baseline as the background spectrum. The total organic carbon (TOC) concentrations were measured on a TOC analyzer (TOC-L-CPH, Shimadzu, Japan) by high-temperature catalytic oxidation. The light-absorbing properties were quantified via the calculation of the mass absorption efficiency at 365 nm (MAE<sub>365</sub>, m<sup>2</sup> g<sup>-1</sup>C) and through Eq. (3) (You et al., 2020; Tang et al., 2022b).

$$MAE_{\lambda} = \frac{A_{\lambda} \times ln(10)}{C_{mass} \times l} \tag{3}$$

In Eq. (3),  $A_{\lambda}$  is the UV-vis absorbances,  $C_{mass}$  is the TOC concentration measured in g m<sup>-3</sup>, and l is the path length (0.01 m). Fluorescence excitation-emission matrix (EEM) of the samples collected at the initial time (t = 0) and the end time (t = 5 h) of photooxidation reactions were obtained by a fluorescence spectrophotometer (F-4700, Hitachi, Japan) in the three-dimensional scan mode at room temperature. The instrument conditions were set as the following parameters: excitation and emission wavelength ranges were 200-600 and 220-700 nm; scanning intervals were 5 and 2 nm; slit widths were 10 nm; and scan speeds were 30000 nm min<sup>-1</sup>. The fluorescence data is calibrated by a previous study prior to analysis of results. It includes instrumental bias correction, inner filter effect correction, removal of Raman and Rayleigh scattering, and blank subtraction (Murphy et al., 2013).

## 2.6 Measurements of chemical composition of products

Inorganic products of the samples collected at time intervals of one hour were quantified via ion chromatography (IC, Dionex ICS-6000, Thermo Scientific, USA) using an anion column (Dionex IonPacTM AS15, 4 mm ID × 250 mm, Thermo Scientific, USA) with 38 mM KOH as an eluent at a flow rate of 1.2 mL min<sup>-1</sup>. The sample was filtered through a microporous membrane syringe filter (PTFE, pore size: 0.22 µm) before the IC measurement.

Organic products of the samples collected at the end time (t = 5 h) of photooxidation reactions were carried out with LC (Vanquish, Thermo Scientific, USA) coupled with the Orbitrap mass spectrometer (Orbitrap Exploris 480, Thermo Scientific, USA) equipped with the ESI probe, and the full description of the LC-Orbitrap MS method has been provided in the Supplement (Sec. S2). The Xcalibur and Compound Discoverer 3.3 software (Thermo Fisher Scientific Inc., USA) were used to analyze the data from Orbitrap MS. Nanoparticle tracking analysis (NTA, Nanosight NS300, Malvern Panalytical,

UK) was used for determination of number concentration and size distribution profile of nanoparticles in the samples collected at time intervals of one hour.

## 3 Results and discussion

### 3.1 Rate constants for reactions of BTs with OH radicals

Rate constants of BT, MBT, and CBT reacted with OH radicals were obtained at initial pH 2 and 10 via the relative rate method. The slope of the fitted straight line obtained by plotting  $\ln([BTs]_0/[BTs]_t)$  versus  $\ln([Ref]_0/[Ref]_t)$  is equal to  $k_{RT} / k_{Ref}$ . In Fig. 2, the relative kinetic plots obtained were linear ( $R^2 > 0.99$ ), and their intercepts were very close to zero, which demonstrates negligible contribution of secondary reactions (Wang et al., 2015). The fitted slopes in Fig. 1 were used to calculate the rate constants by multiplying the slopes by rate constants of reference compounds. The determined second-190 order rate constants of BT, MBT, and CBT reacted with OH radicals are listed in Table 2. The rate constants of BT + OH, MBT + OH, and CBT + OH reactions were determined to be  $(8.0 \pm 1.8)$ ,  $(7.6 \pm 1.7)$ , and  $(7.6 \pm 1.9) \times 10^9$  M<sup>-1</sup> s<sup>-1</sup> at initial pH 2, and  $(9.7 \pm 2.7)$ ,  $(9.8 \pm 2.7)$ , and  $(9.4 \pm 2.7) \times 10^9$  M<sup>-1</sup> s<sup>-1</sup> at initial pH 10. The rate constants of selected BTs reacted with OH radicals under the highly acidic condition were slightly lower than those under the weakly alkaline condition. The values of BT, MBT, and CBT were very close under each initial pH condition. The rate constants for the BTs determined in 195 this study are typical for aromatic contaminants (Li et al., 2023b). The second-order rate constant of BT reacted with OH radicals were reported previously, which was determined to be  $(8.61 \pm 0.23) \times 10^9 \,\mathrm{M}^{-1} \,\mathrm{s}^{-1}$  at pH 7 and room temperature and consistent with the value in this study (Bahnmüller et al., 2015). However, some previous studies reported lower rate constants of the BT + OH reaction in the aqueous phase, which were  $9.5 \times 10^8$  and  $3.85 \times 10^9$  M<sup>-1</sup> s<sup>-1</sup> at pH 7 (Borowska et al., 2016; Andreozzi et al., 2001). The lower values from previous studies are potentially due to the differences in 200 experimental conditions. The reaction temperatures in the previous studies were 293 K, lower than that in our study. In addition, the material of vessel employed in the previous studies is ordinary glass, but quartz glass in our study. The reduction in UV light caused by ordinary glass, together with the higher initial concentration of BT, limits the availability of OH radicals. These factors thus potentially yield lower measured rate constants. Overall, the aqueous-phase kinetic result indicates a high reactivity of BTs with OH radicals, suggesting that atmospheric aqueous-phase oxidation could be a 205 significantly effective process to degrade these contaminants from the air.

Figure 2. Relative kinetic plots for oxidation of BT with OH radicals in the aqueous phase at initial pH 2 and 10 using SA, LPA, and TA as the reference compounds ([BTs]<sub>0</sub>:[Refs]<sub>0</sub> = 1:1).

Table 2. Kinetics data for BT, MBT, and CBT + OH reactions.

| Reactant | pН | Reference | Number of runs | $k (\times 10^9 \mathrm{M}^{\text{-}1} \mathrm{s}^{\text{-}1})$ |
|----------|----|-----------|----------------|-----------------------------------------------------------------|
| BT       | 2  | SA        | 3              | $8.0 \pm 1.8$                                                   |
|          |    | LPA       | 3              |                                                                 |
|          | 10 | TA        | 3              | $9.7\pm2.7$                                                     |
|          |    | LPA       | 3              |                                                                 |
| MBT      | 2  | SA        | 3              | $7.6 \pm 1.7$                                                   |
|          |    | LPA       | 3              |                                                                 |
|          | 10 | TA        | 3              | $9.8\pm2.7$                                                     |
|          |    | LPA       | 3              |                                                                 |
| CBT      | 2  | SA        | 3              | $7.6 \pm 1.9$                                                   |
|          |    | LPA       | 3              |                                                                 |
|          | 10 | TA        | 3              | $9.4\pm2.7$                                                     |
|          |    | LPA       | 3              |                                                                 |

## 210 3.2 Optical properties

The aqueous-phase reactions of BT, MBT, and CBT with OH radicals were accompanied by the alteration in the optical properties of the reaction solution. It can be directly observed that the color of the reaction solution gradually changed from transparent to yellowish-brown after 5 hours of UV irradiation (Fig. 3a), which indicated the formation of light-absorbing species from the oxidation process of selected BTs initiated by OH radicals in the aqueous phase. It was consistent with the

215 aqueous-phase oxidation of phenolic compounds in the atmosphere, which have been proved to contribute to the formation of atmospheric brown carbon (Desyaterik et al., 2013; Li et al., 2023b; Teich et al., 2017). The UV-vis absorption spectra of reaction solutions consisting of 0.45 mM BTs and 10 mM H<sub>2</sub>O<sub>2</sub> were recorded. In Fig. 3b and S3, the principal absorbance band centered closely as the starting compound and the initial pH differed, which was in the range of 251-256 nm. The principal absorbance band underwent a rapid decrease as the reaction processed. The absorbance bands at 251-256 nm 220 agreed with the characteristic peak of BTs. It was ascribed to the  $\pi$ - $\pi$ \* electronic transitions (> 180 nm) in aromatic compounds (Yi et al., 2018; Santos et al., 2016). Simultaneously, the absorption spectra obviously increased in the range from near-UV (320-400 nm) to vis regions, in which the selected BTs did not absorb light. The increase of spectra in this region suggested that the formation of oligomers with large and conjugated  $\pi$ -electron structures (Chang and Thompson, 2010). It could be proposed that aqueous-phase oxidation of BTs initiated by OH radicals led to the degradation of precursor 225 compounds and the formation of oligomeric products. Moreover, it was noting that this process formed products characterized by efficient absorption spectra those smoothly increased from the vis to UV wavelengths, which could contribute to the mass of the atmospheric brown carbon (Laskin et al., 2015).

The UV-vis absorption spectra were converted into mass absorption efficiency (MAE) using Eq. (3). After 5 h of photooxidation, the mass absorption efficiency at 365 nm (MAE<sub>365</sub>) values of the light-absorbing substances derived from BTs reached 0.76, 0.74, 0.34, 0.50, 0.85, and 1.02 m<sup>2</sup> g<sup>-1</sup> for Exps. BT-pH2, BT-pH10, MBT-pH2, MBT-pH10, CBT-pH2, and CBT-pH10, respectively (Fig. 3c and Fig. S4). Among them, the products from CBT exhibit the highest MAE<sub>365</sub>, resulting from the higher electronegativity of the -Cl group in CBT. MAE<sub>365</sub> values of the selected BTs-driven products are comparable to those reported for laboratory-generated brown carbon formed via Maillard-like aqueous-phase reactions from carbonyl compounds mixed with ammonium sulfate or amine, where MAE<sub>365</sub> ranged from 0.15 to 4.50 m<sup>2</sup> g<sup>-1</sup> depending on precursor combinations (Tang et al., 2022b). They are also within the MAE<sub>365</sub> range observed in ambient brown carbon samples from biomass and coal combustion sources, which span from 0.21 to 3.10 m<sup>2</sup> g<sup>-1</sup> (Zhang et al., 2024). It suggests that aqueous-phase oxidation of BTs can form light-absorbing organic products with optical properties relevant to atmospheric brown carbon.

Figure 3. Characterization plots of the reaction solution in Exp. BT-pH2. The change of solution color (a). UV-vis absorption spectra (b) and Mass absorption efficiency (MAE) (c) of reaction solution collected at reaction time intervals of 1 h. The change of EEM fluorescence spectra (d). Time profiles of BT degradation, inorganic products formation (SO4<sup>2-</sup>, NO3<sup>-</sup>, and Cl<sup>-</sup>), particles formation, and total organic carbon (TOC) concentration (e). Size distribution of nanoparticles formed at reaction time intervals of 1 h (f). The change of NTA images (g).

Fluorescence has been used recently to analyze water-soluble organics due to its high sensitivity and nondestructive analysis 245 characteristics, Fig. 3d and Fig. S5 show the EEM fluorescence spectra before and after 5-h reaction of BT, MBT, and CBT with OH radicals at initial pH 2 and 10. The fluorescence results of reaction systems herein reveal longer emission and excitation wavelengths after 5 h of photooxidation. Red shifts in both emission and excitation wavelengths are usually related to an increase in the size of the ring system and an increase in the degree of conjugation (Chang and Thompson, 2010). This change in the fluorescence spectra also suggests that the degradation of the initial compound, and the newly 250 formed compounds at longer wavelengths may have a more complex structure than its precursor, probably with the presence of condensed aromatic ring and other  $\pi$ -electron systems, with a high level of conjugation (Chen et al., 2002). Moreover, EEM fluorescence features of BTs-driven products are consistent with the LO-HULIS (less-oxygenated humic-like substances) group, a subclass of atmospheric brown carbon identified in studies of ambient aerosols (Jiang et al., 2022). LO-HULIS are less-oxygenated and nitrogen-containing aromatic compounds commonly observed in biomass burning aerosols. 255 Given the spectral similarity, the aqueous-phase OH oxidation products of BTs in our study can be reasonably classified as LO-HULIS-type chromophores, further indicating a potential for contributing to the light absorption of aerosols.

## 3.3 Formation of nanoparticles





The reactions of BT, MBT, and CBT with OH radicals at initial pH 2 and 10 in the agueous phase can lead to the formation of nanoparticles. Fig. 3g and S7 shows NTA images of samples collected before and after 5 hours of photooxidation of BT, MBT, and CBT with OH radicals at initial pH 2 and 10. It can be obviously observed that there are rarely particles in a NTA view from the samples collected before irradiation, but the particles are widely present after 5 hours of photooxidation in all reaction solution samples. In Fig. 3f and S8, NTA shows that the particles formed from photooxidation of the selected BTs are in the nanometer scale, with size distributions ranging from 50 to 400 nm. In Fig. 3e and S6, the number concentration of the nanoparticles after 5 hours of photooxidation were on the order of 10<sup>8</sup> particles mL<sup>-1</sup>, much greater than those before irradiation. The size distribution shows large polydispersity, with individual particles varying in size from a few to several hundred nanometers. While previous studies have reported the formation of oligomeric and polymeric products in atmospheric relevant aqueous-phase reactions (Li et al., 2023a; Li et al., 2023b; Tang et al., 2022b), direct detection of newly formed nanoparticles remains rare. The observed nanoparticles may originate from the aggregation of oligomeric products from aqueous-phase BTs oxidation, highlighting a potentially important but underrecognized route of secondary aerosols formation from aqueous-phase chemistry. It is noteworthy that NTA provides only an approximate estimation of nanoparticles within a particle size range of 10-2000 nm. Additionally, nanoparticles may undergo agglomeration or deagglomeration and may partially dissolve during the transfer from the reaction solution. It can significantly impact particle concentrations and size distributions. Therefore, it should be aware that the particle size distributions may not be precisely reflected.

## 3.4 Products analysis and mechanisms







IC measurements were performed for the analysis of inorganic products formed from the photooxidation of BT, MBT, and CBT with OH radicals in the aqueous phase. In BT and MBT experiments inorganic products contain SO<sub>4</sub><sup>2-</sup> and NO<sub>3</sub>-, and in CBT experiments they contain SO<sub>4</sub><sup>2-</sup>, NO<sub>3</sub><sup>-</sup>, and Cl<sup>-</sup>. Fig. 3e and S6 show that in the selected BTs photooxidations, inorganic products gradually generate with the degradation of parent compounds. But the formation yields of inorganic products are different regardless of the parent compounds. The molar yields of SO<sub>4</sub><sup>2</sup>- are in the range of 19.4-46.6%, significantly higher than those of NO<sub>3</sub>, which falls within 10.1% in the selected BTs experiments (Table S3). It is worth noting that in the CBT experiments the molar yields of Cl<sup>-</sup> are the highest among the inorganic products, which are in the range of 61.5-87.1%. The analysis of inorganic products in the reaction solutions indicates that aqueous-phase oxidation of BTs has the capacity to form sulfate in the atmosphere, and that nitrate and chloride can be generated from specific BTs. It possibly suggests the presence of inorganic aerosol produced during aqueous-phase BT oxidation in the atmosphere. In a previous study, a laboratory simulation study of gas-phase BT oxidation in the atmosphere was conducted using a Potential Aerosol Mass Oxidation Flow Reactor (PAM-OFR). The result indicates that gas-phase BT oxidation has the capacity to form sulfur dioxide and sulfuric acid in the atmosphere, possibly suggesting the generation of sulfate aerosol during BT oxidation (Franklin et al., 2021). Considering both aqueous- and gas-phase oxidation results, atmospheric BTs oxidation appears to be a potential source of sulfate aerosols, given the significant sulfate yields observed from the oxidation of sulfur-containing structures.

LC-ESI-Orbitrap MS measurements were performed in both positive- and negative-ion ESI modes for the analysis of organic products formed from the photooxidation of BT, MBT, and CBT with OH radicals in the aqueous phase. The raw data from LC-ESI-Orbitrap MS were processed with deconvolution, aligning retention times, subtraction of background compounds, and composition prediction. The number of atoms ( $^{12}C \le 30$ ,  $^{1}H \le 30$ ,  $^{16}O \le 20$ ,  $^{14}N \le 10$ ,  $^{32}S \le 10$  and  $^{35}Cl \le 5$ ) and a constraint of 4 ppm mass tolerance are applied as constraints to obtain the elemental composition. Many organic products with assigned molecular formulas and certain retention times were ultimately identified. Fig. 4a shows that organic products formed by each selected BTs photooxidation contain hundreds of identified compounds in both ESI modes. It suggests that the aqueous-phase oxidation of BTs has the potential to considerably enhance the diversity of chemical composition in the atmosphere. Meanwhile, products contained more identified compounds in positive-ion (+ESI) mode than in negative-ion (-ESI) mode, which suggests a possible enrichment of basic functionalities in the product mixture, as such groups are generally more amenable to ionization in +ESI mode (Lin et al., 2012). The numbers of identified compounds between +ESI and -ESI modes are less different, and the identified compounds across different carbon atom numbers (nC) in both modes are irregular. It indicates that a combined consideration of +ESI and -ESI modes is necessary for products analysis rather than a single mode. It can be also observed from Fig. 4a that the number of identified compounds shows obviously periodic fluctuation with the nC. Four crest values can be observed from the BT experiments, which are at nC 7, 13-14, 21-22, and 27-28. The nC difference value is in the vicinity of 7, which is the nC of BT molecular formula. Similar patterns can be observed from MBT and CBT experiments. It suggests the generation of oligomers with the nC of parent compounds as the mainly repetitive units, which probably contribute to the formation of nanoparticles in the aqueous phase photooxidation reactions of the selected BTs with OH radicals.

Figure 4. Plots of blank-corrected Orbitrap mass spectra of 5 hours of photooxidation of BT, MBT, and CBT with OH radicals at initial pH 2 and 10 in negative- (-ESI) and positive- (+ESI) ion modes. The total number of assigned formulas as a function of the number of carbon atoms (a). The percentages of number distribution of CHO, CHON, CHOS, CHONS, CHOCI, CHONCI, CHOSCI, CHONSCI and other assigned molecular formula (b).

The molecular formula of identified compounds can be classified into CHO, CHON, CHOS, and CHONS groups according to their elementary composition (Tang et al., 2020a). CHONS compounds account for the largest proportion of the overall molecular formula of identified compounds, around 90% in +ESI mode and 80% in -ESI mode for BT and MBT oxidation systems (Fig. 4b). In the CBT experiment, CHONS compounds alone account for only 30-40% of the total identified molecular formulas. However, when CHONSCl compounds are included, the combined proportion of CHONS and CHONSCI compounds becomes comparable to the fraction of CHONS compounds identified in the BT and MBT experiments. CHONS compounds have been shown to contribute significantly to particle light absorption, with higher mass absorption efficiencies observed in CHONS-rich fractions (Zhang et al., 2024; Bao et al., 2023). With the formation of abundant CHONS species, the aqueous-phase BTs oxidation may indicate a potential contribution to the climate-relevant properties of brown carbon through enhanced light absorption. In addition, CHON compounds in the selected BT experiments (CHON + CHONCl compounds in CBT experiments) account for ~10% of the overall molecular formula of identified compounds, significantly higher than the CHOS compounds (CHOS + CHOSCl compounds in CBT experiments). It is consistent with the lower molar yields of NO<sub>3</sub><sup>-</sup> than those of SO<sub>4</sub><sup>2</sup>- formation in the inorganic products measurements. In CBT experiments the proportion of organic products containing Cl atom is lower than that of organic products containing S and N atoms, which is consistent with the highest molar yields of Cl<sup>-</sup>. It should be noted that in this study the LC-Orbitrap MS analysis provides more qualitative insights, as quantitative determination was limited and not conducted. It can be considered and developed for further characterization of the formation and evolution of organic products.








The structures of BT and proposed products, and pathways in the reaction of BT with OH radicals are shown in Fig. 5. The N, C5, and C7 atoms have relatively lower charge distribution and high electron density, which should be the reactive sites for electrophilic radicals attack (Zhou et al., 2019). OH radicals can electrophilically attack the heterocycle ring or the benzene ring of BT in the first step, leading to the H abstraction and formation of Radicals (I) or (II), respectively (Fig. 5a). Then, the organic radicals would further react with OH radicals, forming hydroxylated BT. It is worth paying attention to 2hydroxybenzothiazole (2OBT) formed from Radical (I). Liao et al explored six PM<sub>2.5</sub>-bound BTs in three typical Chinese cities and found that OBT was always predominant, accounting for 50-80% of total BTs (Liao et al., 2021). The hydroxylated BT formed from Radical (II) has four isomers featuring a hydroxy group at C4-7 positions. Hydroxylated BTs can be further attacked by OH radicals to form hydroxylated radicals. Tens of dihydroxylated and trihydroxylated BTs can be generated and measured in mass spectra. Zhou et al studied the removal of BT in persulfate promoted wet air oxidation (PWAO) and found hydroxylated and dihydroxylated BTs using an ultra-performance liquid chromatography quadrupole time-of-flight mass spectrometry (UPLC-QTOF-ESI-MS), but trihydroxylated BTs were not measured (Zhou et al., 2019). It was proposed that the UV/persulfate degradation of benzothiazole could result in the formation of dihydroxylated and trihydroxylated BTs based on the UPLC-QTOF-ESI-MS, but hydroxylated BTs were not mentioned (Lai et al., 2023). 2OBT tends to undergo heterocyclic ring cleavage by C2-S bond broken as this bond has the largest bond length in the molecule that is much easier to get broken (Fig. 5b) (Zhou et al., 2019). A series of chemical components obtained from mass spectra can verify this reaction pathway. Following the breakage of the heterocyclic ring, a batch of carboxylic acids with a benzene ring are generated step by step accompanied by the formation of inorganic products, e. g. SO<sub>4</sub><sup>2-</sup> and NO<sub>3</sub><sup>-</sup>. This process is along with a decrease of the nC in products. Then, the benzene moiety goes through a ring-opening process, leading to a further decrease of nC. The process of ring-opening of thiazole and benzene moieties leads to the formation of products with nC less than the parent compound BT. Fig. 5c shows an example of the fragmentation of dihydroxy BTs, which is similar to the fragmentation pathway of 2OBT. The benzene moiety of dihydroxy BTs can undergo the carbonylation process, leading to the formation of quinone compounds. The ring-opening process takes place, which leads to the formation of dicarboxylic acids and an nC decrease in products.

Moreover, multifunctional oligomers can be generated from the reaction of BT and OH radicals in the aqueous phase. Radical (I) can undergo C-C coupling with one another at the C7 position, resulting in the formation of an isomeric form of the dimer C<sub>14</sub>H<sub>8</sub>NS (Fig. 5d). Additionally, C-C coupling can take place between a Radical (I) and a Radical (II), or between two Radicals (I), leading to the generation of further isomeric forms of the dimer C<sub>14</sub>H<sub>8</sub>NS. Then the dimers further react with OH radicals, forming polyhydroxylated dimers C<sub>14</sub>H<sub>8</sub>NSO<sub>1-6</sub> (Fig. 5e). BT dimers can react with Radicals (I) or (II) to form BT trimers with a batch of isomers. Then polyhydroxylated trimers can be generated. It should be noted that Fig. 5 only illustrates some examples of the reaction pathways between BT and OH radicals for the purpose of providing clear descriptions. As a matter of fact, processes of hydroxylation, carbonylation, fragmentation, oligomerization, and ring-opening have combined effects in the reaction system, ultimately leading to the formation of hundreds of multifunctional products with a large range of nC from a parent compound. It is reasonable to speculate that multifunctional oligomers may contribute to the formation of nanoparticles from aqueous-phase oxidation of BT by OH radicals.

Figure 5. Proposed reaction mechanism and structures of partial products of BT oxidized by OH radicals in the aqueous phase. Red structures with formulas and molecular weight are the products assigned in Orbitrap mass spectra.

## 4. Conclusions






The atmospheric aqueous-phase lifetime ( $\tau$ ) of BT, MBT, and CBT initiated by OH radicals can be evaluated by Eq. (4).  $\tau = 1/(k_{BT_S} \times \text{[OH]})$ 

In Eq. (4),  $k_{BT}$ , is the rate constants of BT, MBT, and CBT reacted with OH radicals determined in this study, and [OH] represents the concentration of OH radicals in the atmosphere under various aqueous-phase conditions (Bianco et al., 2020; Arakaki et al., 2013; Herrmann et al., 2010). The aqueous-phase OH concentrations and the evaluated lifetimes of BT, MBT, and CBT are summarized in Table 3. Atmospheric aqueous phases are generally categorized into cloud droplets and deliquescent particles, and can be further classified by location as urban, remote, and maritime regions. Previous studies give the mean value of OH concentrations in various aqueous phases (Bianco et al., 2020; Hermann et al., 2010). The atmospheric lifetimes determined for BT, MBT, and CBT are in the range from 0.01 to 0.4 h in deliquescent particles, and 0.01 to 10.4 h in cloud particles. Previous studies also model varying ranges of OH concentrations in each aqueous phase, from 102 to 104 fold (Bianco et al., 2020; Arakaki et al., 2013; Hermann et al., 2010). For example, the OH concentrations are from 10<sup>-14</sup> to 10<sup>-12</sup> M in remote deliquescent particles and 10<sup>-16</sup> to 10<sup>-12</sup> M in urban deliquescent particles. The 10<sup>2</sup> to 10<sup>4</sup>-fold ranges in aqueous-phase OH concentrations produce 10<sup>2</sup> to 10<sup>4</sup>-fold spread in estimated BTs lifetimes based on Eq. (4). The large range of OH concentrations leads to large uncertainties of estimated BTs lifetimes. In urban regions, the OH concentrations are modeled to be from 10<sup>-16</sup> to 10<sup>-12</sup> M in both cloud droplets and deliquescent particles, which leads to considerably large ranges of BTs lifetimes, from several minutes to several days. In addition to this, in most cases, the BTs lifetimes can be limited to several minutes to several hours. In general, the selected BTs tend to be transformed by aqueous OH radicals, especially in the region of their emission sources, and in rare cases they are also probably more persistent in the aqueous phase. It should be noted that the pH conditions in this study may introduce additional uncertainty into the estimated atmospheric lifetimes of BTs, due to the pH-dependent variability in aqueous-phase OH radical concentrations (Wolke et al., 2005; Herrmann et al., 2005).

Table 3. Atmospheric lifetimes of BT, MBT, and CBT initiated by OH radicals in various aqueous and gas phases.

|                       | Ref.                                             |      | [OH] <sup>a</sup>     | τ <sub>BT</sub> in | n h <sup>b</sup> | $	au_{	ext{MBT}}$ in | h <sup>b</sup> | τ <sub>CBT</sub> ii | n h <sup>b</sup> |
|-----------------------|--------------------------------------------------|------|-----------------------|--------------------|------------------|----------------------|----------------|---------------------|------------------|
| *                     | Herrmann et al. (2010)<br>& Bianco et al. (2020) | Mean | $3.5 \times 10^{-15}$ | 9.9                | 8.2              | 10.4                 | 8.1            | 10.4                | 8.4              |
|                       |                                                  | Max  | $1.6 \times 10^{-14}$ | 2.2                | 1.8              | 2.3                  | 1.8            | 2.3                 | 1.8              |
|                       |                                                  | Min  | $2.9 \times 10^{-16}$ | 119.7              | 98.7             | 126.0                | 97.7           | 126.0               | 101.9            |
|                       | Arakaki et al. (2013)                            | Max  | $1.9 \times 10^{-12}$ | 0.02               | 0.02             | 0.02                 | 0.01           | 0.02                | 0.02             |
|                       |                                                  | Min  | $1.0 \times 10^{-14}$ | 3.5                | 2.9              | 3.7                  | 2.8            | 3.7                 | 3.0              |
| Remote cloud droplets | Herrmann et al. (2010)                           | Mean | $2.2 \times 10^{-14}$ | 1.6                | 1.3              | 1.7                  | 1.3            | 1.7                 | 1.3              |

|                                 | & Bianco et al. (2020)                  | Max     | $6.9\times10^{\text{-}14}$ | 0.5   | 0.4                | 0.5   | 0.4   | 0.5   | 0.4   |
|---------------------------------|-----------------------------------------|---------|----------------------------|-------|--------------------|-------|-------|-------|-------|
|                                 |                                         | Min     | $4.8 \times 10^{-15}$      | 7.2   | 6.0                | 7.6   | 5.9   | 7.6   | 6.2   |
|                                 | Arakaki et al. (2013)                   | Max     | 2.4 × 10 <sup>-12</sup>    | 0.01  | 0.01               | 0.02  | 0.01  | 0.02  | 0.01  |
|                                 |                                         | Min     | $2.6 \times 10^{-14}$      | 1.3   | 1.1                | 1.4   | 1.1   | 1.4   | 1.1   |
| Maritime cloud droplets         | Herrmann et al. (2010)                  | Mean    | 2.0 × 10 <sup>-12</sup>    | 0.02  | 0.01               | 0.02  | 0.01  | 0.02  | 0.01  |
|                                 | & Bianco et al. (2020)                  | Max     | $5.3 \times 10^{-12}$      | 0.01  | 0.01               | 0.01  | 0.01  | 0.01  | 0.01  |
|                                 |                                         | Min     | $3.8 \times 10^{-14}$      | 0.9   | 0.8                | 1.0   | 0.7   | 1.0   | 0.8   |
|                                 | Arakaki et al. (2013)                   | Max     | $2.0 \times 10^{-12}$      | 0.02  | 0.01               | 0.02  | 0.01  | 0.02  | 0.01  |
|                                 |                                         | Min     | $1.8 \times 10^{-13}$      | 0.2   | 0.2                | 0.2   | 0.2   | 0.2   | 0.2   |
| Urban deliquescent              | Herrmann et al. (2010)                  | Mean    | $4.4 \times 10^{-13}$      | 0.1   | 0.1                | 0.1   | 0.1   | 0.1   | 0.1   |
| particles                       |                                         | Max     | $1.9 \times 10^{-12}$      | 0.02  | 0.02               | 0.02  | 0.01  | 0.02  | 0.02  |
|                                 |                                         | Min     | $1.4\times10^{\text{-}16}$ | 248.0 | 204.5              | 261.1 | 202.5 | 261.1 | 211.1 |
|                                 | Arakaki et al. (2013)                   |         | $8.0 \times 10^{-13}$      | 0.04  | 0.04               | 0.05  | 0.04  | 0.05  | 0.04  |
|                                 | Herrmann et al. (2010)                  | Mean    | $3.0 \times 10^{-12}$      | 0.01  | 0.01               | 0.01  | 0.01  | 0.01  | 0.01  |
| particles                       |                                         | Max     | $8.0\times10^{-12}$        | 0.004 | 0.004              | 0.005 | 0.004 | 0.005 | 0.004 |
|                                 |                                         | Min     | $5.5\times10^{-14}$        | 0.6   | 0.5                | 0.7   | 0.5   | 0.7   | 0.5   |
|                                 | Arakaki et al. (2013)                   |         | $3.6 \times 10^{-12}$      | 0.01  | 0.01               | 0.01  | 0.01  | 0.01  | 0.01  |
| Maritime deliquescent particles | Herrmann et al. (2010)                  | Mean    | $1.0 \times 10^{-13}$      | 0.3   | 0.3                | 0.4   | 0.3   | 0.4   | 0.3   |
|                                 |                                         | Max     | $3.3 \times 10^{-12}$      | 0.01  | 0.01               | 0.01  | 0.01  | 0.01  | 0.01  |
|                                 |                                         | Min     | $4.6 \times 10^{-15}$      | 7.5   | 6.2                | 7.9   | 6.2   | 7.9   | 6.4   |
|                                 | Arakaki et al. (2013)                   |         | $4.0 \times 10^{-16}$      | 86.8  | 71.6               | 91.4  | 70.9  | 91.4  | 73.9  |
| Gas phase                       | Prinn et al. (2001)                     | Mean    | $1.0 \times 10^{6}$        | 39.7° | 132.3 <sup>d</sup> | 15.1° |       | 56.4° |       |
|                                 | Finlayson-Pitts and Pitts<br>Jr. (2000) | Mid-day | $1.0 \times 10^{7}$        | 4.0°  | 13.2 <sup>d</sup>  | 1.5°  |       | 5.6°  |       |
|                                 |                                         | 2 .     |                            |       |                    |       |       |       |       |

<sup>&</sup>lt;sup>a</sup>The units of OH concentrations are M and molecule cm<sup>-3</sup> in the aqueous and gas phases, respectively.

<sup>&</sup>lt;sup>b</sup>The two lifetimes in the aqueous phase are calculated by the rate constants at pH 2 and 10 obtained in this study.

<sup>&</sup>lt;sup>c</sup>The lifetimes in the gas phase are calculated based on the rate constants from the Atmospheric Oxidation Program for Microsoft Windows (AOPWIN) model, which are 0.7, 1.8, and 0.5 × 10<sup>-11</sup> cm<sup>3</sup> molecule<sup>-1</sup> s<sup>-1</sup> for reactions of OH with BT, 400 MBT, and CBT, respectively (USEPA, 2012).

<sup>&</sup>lt;sup>d</sup>The lifetimes of BT in the gas phase are calculated by the rate constants determined by a relative rate method, which is  $(2.1 \pm 0.1) \times 10^{-12}$  cm<sup>3</sup> molecule<sup>-1</sup> s<sup>-1</sup> (Karimova et al., 2024).

The atmospheric gas-phase lifetime of BT, MBT, and CBT initiated by OH radicals can also be evaluated by Eq. (4). Rate constants of BT, MBT, and CBT reacted with OH radicals have been estimated using the Atmospheric Oxidation Program for Microsoft Windows (AOPWIN) model, which is 0.7, 1.8, and  $0.5 \times 10^{-11}$  cm<sup>3</sup> molecule<sup>-1</sup> s<sup>-1</sup>, respectively (USEPA, 2012). The AOPWIN model is based upon structure-activity relationship (SAR) methods developed by Atkinson and coworkers, which has been widely used to predict the rate constants for atmospheric gas-phase reactions between organic molecules and OH radicals (Atkinson and Arey, 2003; Atkinson, 1986). A typical peak OH radical concentration at mid-day is 10<sup>7</sup> molecules cm<sup>-3</sup>, and a 24-h average OH radical concentration is 10<sup>6</sup> molecules cm<sup>-3</sup> (Prinn et al., 2001; Finlavson-Pitts and Pitts Jr., 2000). Table 3 also summarizes the gas-phase lifetimes of the selected BTs in the atmosphere, which falls in the range of 1.5-5.6 h at  $1.0 \times 10^7$  OH cm<sup>-3</sup> and 15-56 h at  $1.0 \times 10^6$  OH cm<sup>-3</sup>. In addition, the rate constant of the BT + OH reaction has been experimentally determined using the relative rate method, which is  $(2.1 \pm 0.1) \times 10^{-12}$  cm<sup>3</sup> molecule<sup>-1</sup> s<sup>-1</sup> (Karimova et al., 2024). The gas-phase lifetime of BT is translated to be ~13 hours to 5.5 days over the two OH concentrations, which is more than three times higher than that from the AOPWIN model. The lifetimes of BTs estimated using mean OH concentrations in the gas phase are significantly longer than those estimated using mean OH concentrations in the aqueous phase, although in rare cases where aqueous-phase OH concentrations are on the order of 10<sup>-16</sup> M, the estimated lifetimes in the aqueous phase can exceed those in the gas phase by several times. It reveals the high reactivity of BTs with OH radicals in the aqueous phase, suggesting that atmospheric aqueous-phase oxidation could be a significantly effective process to transform these contaminants in the atmosphere.








Significant progress has been made in understanding the formation mechanisms of secondary aerosols, but it is still not possible to quantitatively explain the observed secondary aerosols (Johnston and Kerecman, 2019; Li et al., 2017). This implies the existence of missing sources of precursors and/or unknown atmospheric oxidation mechanisms of known precursors (Goldstein and Galbally, 2007; Wang et al., 2013). In this study, the formation of oxidized organic products with both higher and lower molecular weights was confirmed by LC-Orbitrap MS analysis, and a higher molar yield of sulfate was quantified by IC measurement. Based on these results, aqueous-phase oxidation mechanisms of BTs were proposed, in which OH radicals are assumed to initiate attack on the benzothiazole ring, forming radical intermediates that are subjected to radical-radical coupling, fragmentation, and further oxidation. It suggests that aqueous-phase BTs oxidation has the capacity to contribute to secondary aerosols, including both inorganic and organic components in the atmosphere. The contribution of secondary aerosols generated from BTs is considered non-negligible and can contribute to the fine particulate matters in some regions where BTs concentrations are comparable to common aromatic compounds. In addition, yellowish solutions those absorb from the vis to UV wavelengths and has unusual fluorescence spectra are observed after the selected BTs photooxidation, which suggests that aqueous-phase BTs oxidation might contribute to the mass of the atmospheric brown carbon. The optical properties might be due to formation of high-molecular-weight organic products by radicalradical oligomerization, which is consistent with the measurement result of nanoparticles and confirmed by the LC-Orbitrap MS analysis. BTs aqueous reactions have the potential to contribute to the fine particulate air pollution and might significantly modify the chemical compositions and optical properties of atmospheric particles in regions influenced by BTs emissions and further impact the climate and human health. In the future, research should focus on the occurrence and distribution of BTs in atmospheric aqueous phases, with particular emphasis on integrating real-world BT concentrations and varying ambient conditions to advance the understanding of their atmospheric chemistry. Future research should also focus on selection of key secondary products from BTs oxidation in the ambient atmosphere and their toxicological assessments to evaluate the potential impacts of BTs oxidation on human health.

## **Author contributions**



QZ: Conceptualization, Data curation, Methodology, Investigation, Software, Visualization, Writing - original draft preparation, Funding acquisition. WZ: Methodology, Investigation. SST: Conceptualization, Methodology. KH: Visualization, Methodology. JF: Methodology. ZCY: Conceptualization, Methodology. YHT: Investigation. SYS: Visualization. YM: Investigation. XZY: Investigation. JJF: Writing - review & editing, Supervision, Resources, Project administration, Funding acquisition. GBJ: Supervision.

## **Competing interests**

The contact author has declared that none of the authors has any competing interests.

## 450 Data availability

All data are available from the authors upon request by contacting Jianjie Fu (jifu@rcees.ac.cn).

### Acknowledgments

The authors gratefully acknowledge the National Natural Science Foundation of China (22406037, 22376044, and 22476006), the Strategic Priority Research Program of the Chinese Academy of Sciences (XDB0750100), the Chinese Academy of Sciences Project for the Youth Innovation Promotion Association (2022022), the Young Scientists in Basic Research (YSBR-086), and the Postdoctoral Science Foundation of Hangzhou, China (E2BH2B0604).

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
