# Peer review of "Potential contribution to secondary aerosols from benzothiazoles in the atmospheric aqueous phase based on oxidation and oligomerization mechanisms"

_EGUsphere, 2025_

## Author Comment (AC1)

**Author's Response to Referee #1**

We greatly appreciate the time and effort that Referee #1 has devoted to reviewing our manuscript. The comments are thoughtful and helpful in improving the quality of our paper. Below we make a point-by-point response to these comments. The response to Referee #1 is structured in the following sequence: (1) comments from the referee in blue color, (2) our response in black color, and (3) our changes in the revised manuscript in red color.

This study investigates the aqueous-phase oxidation kinetics and oligomerization pathways of benzothiazoles (BTs), with notable atmospheric presence but poorly understood liquid-phase reactivity. By experimentally determining OH-radical reaction rate constants for BT, MBT, and CBT, the work provides quantitative data to constrain their atmospheric lifetimes in cloud droplets and aerosol liquid water, an underexplored aspect of BTs' environmental fate. The identification of oligomers via Orbitrap MS, coupled with nanoparticle formation observed through NTA, offers mechanistic evidence for BTs' contribution to secondary organic aerosols (SOA) through aqueous-phase processing. These findings complement existing knowledge on gas-phase BT oxidation and highlight the need to account for aqueous reactions in models predicting SOA formation in urban and industrial regions. The study advances the mechanistic understanding of heterocyclic SVOC transformations, particularly for sulfur- and nitrogen-containing species, and provides a foundation for future investigations into their role in brown carbon formation and regional haze events. These results can potentially be of interest to the ACP audiences. However, several minor issues remain in the present study and should be addressed with additional explanations and revisions. I suggest a minor revision for this manuscript.

1. In Sec. 1: The manuscript would benefit from a more narrative-driven introduction that sets up the central question: What are we missing in our current models of SOA formation? Rather than solely listing prior studies, characterizing BTs as a "missing piece" in the urban secondary aerosol puzzle could provide a sharper hook.

**Response:** We thank the reviewer for this thoughtful suggestion. A more narrative-driven introduction can help clarify the central motivation of the study. We have modified and revised a corresponding sentence in the third paragraph of the Introduction to emphasize the role of BTs as a potentially "missing piece" in current models of SOA formation, as the reviewer suggested. Our modification can better highlight the significance of investigating BTs in the context of urban secondary aerosols formation. Specifically, we now characterize BTs as a potentially important and previously underexplored contributor to secondary aerosols. The revised sentence is as follows: "Therefore, understanding the photooxidation reactions of BTs in the aqueous phase is significant not only for the integrated determination of the atmospheric fate of BTs, but also for assessing their potential contributions to both organic and inorganic components of secondary aerosols due to their nature as sulfur-containing organic compounds."

2. In Line 14: "aeras with heavy traffic" → Correct to "areas with heavy traffic."

**Response:** Taking the reviewer's advice, we have corrected this mistake.

3. In Line 24: The use of the term "illustration" in the abstract should be corrected. Also, the phrase "after illustration" in the Abstract is confusing – the rest of the paper implies it should be simply "after the reaction." Ensure that such terminology is consistent and clear across several sections.

**Response:** Taking the reviewer's advice, we have revised the terminology in the Abstract. "In addition, higher yields of sulfate are formed after the reactions."

4. In Line 145: The authors briefly introduce the calculation method of OH concentrations. I think it would be nice to include a bit more information on the method to estimate OH radical concentration (direct calculation from power output or using chemical actinometry?). Publications from Anastasio group (such as Atmos. Environ., 2015, 100, 230-237) can be referred for the estimation.

**Response:** We thank the reviewer for this suggestion. The steady-state OH was estimated in our study by the same method as in the Supplemental Material of the

reference paper (Atmos. Environ., 2015, 100, 230-237). As shown in Figure R1, the VA loss ($k'$) in this system for pH 2 solutions containing 0.45 mM BT and 10 mM $H_2O_2$ was $1.03 \times 10^{-4}$ s$^{-1}$.

$$k' = k_{BT + OH} [OH]$$

From the BT loss ($k'$), we can determine the steady-state concentration of OH radicals by dividing it by the bimolecular rate constant for BT oxidation by OH that has been determined from our kinetic experiments ($k_{BT + OH} = 8.0 \times 10^9$ M$^{-1}$ s$^{-1}$). Substituting the corresponding values, the average steady-state concentrations of OH ([OH]) in the BT-pH2 experiment was $1.28 \times 10^{-14}$ M. Using this method, [OH] were estimated to be 0.64, 1.34, 0.58, 0.72, and $0.53 \times 10^{-14}$ M for BT-pH10, MBT-pH2, MBT-pH10, CBT-pH2, and CBT-pH10, respectively.

[Figure]

**Figure R1**. The BT loss in the aqueous phase under acidic conditions.

Thus, we have modified the related text at Section 2.4 in the revised manuscript as follows:

"The steady-state concentration of OH radicals can be estimated through dividing the BTs loss by the bimolecular rate constant of BTs with OH radicals (Eq. (2); George et al., 2015). Considering the different parent compounds and the rate constants at pH 2 and 10, the steady-state concentration of OH radicals was estimated to be approximately 10-14 M.

$$[OH] = \frac{k\prime}{k_{\text{BTs + OH}}} \qquad (2)$$

**References**

George, K. M.; Ruthenburg, T. C.; Smith, J.; Yu, L.; Zhang, Q.; Anastasio, C.; Dillner, A. M. FT-IR Quantification of the carbonyl functional group in aqueous-phase secondary organic Aerosol from phenols. Atmos. Environ., 100, 230-237 10.1016/j.atmosenv.2014.11.011, 2015."

5. In Line 193: "could be an significantly effective process" → Correct to "could be a significantly effective process"

**Response:** Taking the reviewer's advice, we have corrected this mistake.

6. LC-Orbitrap MS analysis identified hydroxylated products and oligomers. This piece of evidence is good, but rather qualitative. Did the authors perform LC-Orbitrap MS experiments at different time intervals, so that one can use peak areas of certain products to quantitatively/semi-quantitatively compare with the changes of long-wavelength absorption (e.g., at > 300 nm in Fig 3b)?

**Response:** We thank the reviewer for this suggestion. Time-resolved LC-Orbitrap MS analysis combined with UV-vis spectroscopic monitoring would offer a promising approach for quantitatively linking specific product formation with changes in chromophoric properties (e.g., long-wavelength absorption >300 nm). We did not perform LC-Orbitrap MS measurement at different time intervals under the present experimental condition and thus were unable to conduct quantitative or semi-quantitative analysis of product evolution based on peak area trends. We consider this a worthwhile direction for future investigation. Thus, we have explicitly acknowledged the limitation of quantification in the revised manuscript, which has been added in the third paragraph of Section 3.4. "It should be noted that in this study the LC-Orbitrap MS analysis provides more qualitative insights, as quantitative determination was limited and not conducted. It can be considered and developed for further characterization of the formation and evolution of organic products."

7. In Line 288: In this paragraph, the role of organosulfur and organonitrogen compounds deserves more attention. How might the CHONS products influence particle hygroscopicity, longevity, or climate-relevant properties? These aspects are hinted at but not explored in depth.

**Response:** We thank the reviewer for this valuable comment. The potential implications of CHONS compounds those account for the largest proportion of the overall molecular formula of identified compounds need to be explored in depth. Recent studies have demonstrated that CHONS compounds are important contributors to brown carbon, exhibiting strong light-absorbing capabilities. The presence of CHONS compounds has been positively correlated with increased mass absorption efficiency (MAE), particularly in aromatic-rich fractions. These optical characteristics suggest that CHONS species may influence aerosol radiative properties and thus potentially affect climate-relevant processes. In the revised manuscript, we have added a few sentences about discussion of optical properties following the description of CHONS compounds in the third paragraph of Sect. 3.4 and two corresponding references. "CHONS compounds have been shown to contribute significantly to particle light absorption, with higher mass absorption efficiencies observed in CHONS-rich fractions (Zhang et al., 2024; Bao et al., 2023). With the formation of abundant CHONS species, the aqueous-phase BTs oxidation may indicate a potential contribution to the climate-relevant properties of brown carbon through enhanced light absorption.

**References**

Bao, M., Zhang, Y.-L., Cao, F., Hong, Y., Lin, Y.-C., Yu, M., Jiang, H., Cheng, Z., Xu, R., and Yang, X.: Impact of fossil and non-fossil fuel sources on the molecular compositions of water-soluble humic-like substances in $PM_{2.5}$ at a suburban site of Yangtze River Delta, China, Atmos. Chem. Phys., 23, 8305-8324, 10.5194/acp-23-8305-2023, 2023.

Zhang, L., Li, J., Li, Y., Liu, X., Luo, Z., Shen, G., and Tao, S.: Comparison of water-soluble and water-insoluble organic compositions attributing to different light

absorption efficiency between residential coal and biomass burning emissions, Atmos. Chem. Phys., 24, 6323-6337, 10.5194/acp-24-6323-2024, 2024."

8. In Line 340: "BTs" in "kBTs" should be subscript and the Eq. (3) should be italic, consistent with other equations.

**Response:** Taking the reviewer's advice, we have corrected this mistake.

9. Table 3 is quite dense and might be better split or reorganized for readability.

**Response:** We thank the reviewer for this suggestion. The response to the third comment from Referee #3 involves a revision of the atmospheric lifetimes of BTs that are closely related to Table 3. We have reorganized and modified this table accordingly. The revised Table 3 is as follows:

**Table 3. Atmospheric lifetimes of BT, MBT, and CBT initiated by OH radicals in various aqueous and gas phases.**

| | Ref. | | [OH][a] | $\tau_{BT}$ in h[b] | | $\tau_{MBT}$ in h[b] | | $\tau_{CBT}$ in h[b] | |
|---|---|---|---|---|---|---|---|---|---|
| Urban cloud droplets | Herrmann et al. (2010) & Bianco et al. (2020) | Mean | $3.5 \times 10^{-15}$ | 9.9 | 8.2 | 10.4 | 8.1 | 10.4 | 8.4 |
| | | Max | $1.6 \times 10^{-14}$ | 2.2 | 1.8 | 2.3 | 1.8 | 2.3 | 1.8 |
| | | Min | $2.9 \times 10^{-16}$ | 119.7 | 98.7 | 126.0 | 97.7 | 126.0 | 101.9 |
| | Arakaki et al. (2013) | Max | $1.9 \times 10^{-12}$ | 0.02 | 0.02 | 0.02 | 0.01 | 0.02 | 0.02 |
| | | Min | $1.0 \times 10^{-14}$ | 3.5 | 2.9 | 3.7 | 2.8 | 3.7 | 3.0 |
| Remote cloud droplets | Herrmann et al. (2010) & Bianco et al. (2020) | Mean | $2.2 \times 10^{-14}$ | 1.6 | 1.3 | 1.7 | 1.3 | 1.7 | 1.3 |
| | | Max | $6.9 \times 10^{-14}$ | 0.5 | 0.4 | 0.5 | 0.4 | 0.5 | 0.4 |
| | | Min | $4.8 \times 10^{-15}$ | 7.2 | 6.0 | 7.6 | 5.9 | 7.6 | 6.2 |
| | Arakaki et al. (2013) | Max | $2.4 \times 10^{-12}$ | 0.01 | 0.01 | 0.02 | 0.01 | 0.02 | 0.01 |
| | | Min | $2.6 \times 10^{-14}$ | 1.3 | 1.1 | 1.4 | 1.1 | 1.4 | 1.1 |
| Maritime cloud droplets | Herrmann et al. (2010) & Bianco et al. (2020) | Mean | $2.0 \times 10^{-12}$ | 0.02 | 0.01 | 0.02 | 0.01 | 0.02 | 0.01 |
| | | Max | $5.3 \times 10^{-12}$ | 0.01 | 0.01 | 0.01 | 0.01 | 0.01 | 0.01 |
| | | Min | $3.8 \times 10^{-14}$ | 0.9 | 0.8 | 1.0 | 0.7 | 1.0 | 0.8 |
| | Arakaki et al. (2013) | Max | $2.0 \times 10^{-12}$ | 0.02 | 0.01 | 0.02 | 0.01 | 0.02 | 0.01 |
| | | Min | $1.8 \times 10^{-13}$ | 0.2 | 0.2 | 0.2 | 0.2 | 0.2 | 0.2 |
| Urban deliquescent particles | Herrmann et al. (2010) | Mean | $4.4 \times 10^{-13}$ | 0.1 | 0.1 | 0.1 | 0.1 | 0.1 | 0.1 |
| | | Max | $1.9 \times 10^{-12}$ | 0.02 | 0.02 | 0.02 | 0.01 | 0.02 | 0.02 |
| | | Min | $1.4 \times 10^{-16}$ | 248.0 | 204.5 | 261.1 | 202.5 | 261.1 | 211.1 |

| | | | | | | | | | |
|---|---|---|---|---|---|---|---|---|---|
| | Arakaki et al. (2013) | | $8.0 \times 10^{-13}$ | 0.04 | 0.04 | 0.05 | 0.04 | 0.05 | 0.04 |
| Remote deliquescent particles | Herrmann et al. (2010) | Mean | $3.0 \times 10^{-12}$ | 0.01 | 0.01 | 0.01 | 0.01 | 0.01 | 0.01 |
| | | Max | $8.0 \times 10^{-12}$ | 0.004 | 0.004 | 0.005 | 0.004 | 0.005 | 0.004 |
| | | Min | $5.5 \times 10^{-14}$ | 0.6 | 0.5 | 0.7 | 0.5 | 0.7 | 0.5 |
| | Arakaki et al. (2013) | | $3.6 \times 10^{-12}$ | 0.01 | 0.01 | 0.01 | 0.01 | 0.01 | 0.01 |
| Maritime deliquescent particles | Herrmann et al. (2010) | Mean | $1.0 \times 10^{-13}$ | 0.3 | 0.3 | 0.4 | 0.3 | 0.4 | 0.3 |
| | | Max | $3.3 \times 10^{-12}$ | 0.01 | 0.01 | 0.01 | 0.01 | 0.01 | 0.01 |
| | | Min | $4.6 \times 10^{-15}$ | 7.5 | 6.2 | 7.9 | 6.2 | 7.9 | 6.4 |
| | Arakaki et al. (2013) | | $4.0 \times 10^{-16}$ | 86.8 | 71.6 | 91.4 | 70.9 | 91.4 | 73.9 |
| Gas phase | Prinn et al. (2001) | Mean | $1.0 \times 10^{6}$ | 39.7[c] | 132.3[d] | 15.1[c] | | 56.4[c] | |
| | Finlayson-Pitts and Pitts Jr. (2000) | Mid-day | $1.0 \times 10^{7}$ | 4.0[c] | 13.2[d] | 1.5[c] | | 5.6[c] | |

[a]The units of OH concentrations are M and molecule cm$^{-3}$ in the aqueous and gas phases, respectively.

[b]The two lifetimes in the aqueous phase are calculated by the rate constants at pH 2 and 10 obtained in this study.

[c]The lifetimes in the gas phase are calculated based on the rate constants from the Atmospheric Oxidation Program for Microsoft Windows (AOPWIN) model, which are 0.7, 1.8, and $0.5 \times 10^{-11}$ cm$^{3}$ molecule$^{-1}$ s$^{-1}$ for reactions of OH with BT, MBT, and CBT, respectively (USEPA, 2012).

[d]The lifetimes of BT in the gas phase are calculated by the rate constants determined by a relative rate method, which is $(2.1 \pm 0.1) \times 10^{-12}$ cm$^{3}$ molecule$^{-1}$ s$^{-1}$ (Karimova et al., 2024).

---

## Author Comment (AC2)

**Author's Response to Referee #2**

We greatly appreciate the time and effort that Referee #2 has devoted to reviewing our manuscript. The comments are thoughtful and helpful in improving the quality of our paper. Below we make a point-by-point response to these comments. The response to Referee #2 is structured in the following sequence: (1) comments from the referee in blue color, (2) our response in black color, and (3) our changes in the revised manuscript in red color.

This study employs the laboratory simulation to investigate the aqueous-phase oxidation benzothiazoles (BTs), a class of emerging environmental contaminants. The work determines the kinetics, identifies and categorizes diverse oligomers and functionalized products by LC-Orbitrap MS, which are rarely reported in prior studies on BTs. Molecular formula analyses map key pathways, advancing mechanistic insights into heterocyclic pollutant transformations. Periodic carbon-number patterns empirically confirm oligomer formation, a process in SOA generation. With its analytical depth and mechanistic contributions, this paper is recommended for acceptance after minor revisions.

1. Page 1, Line 16: A grammatical issue in the abstract should be corrected, "are unclear" should be "is unclear".

**Response:** Taking the reviewer's advice, we have corrected this mistake.

2. Page 9: Legends in Fig. 3e are too small. It should be improved for better readability.

**Response:** Taking the reviewer's advice, the legends in Fig. 3e have been enlarged in the revised manuscript as follows:

[Figure]

**Figure 3. Characterization plots of the reaction solution in Exp. BT-pH2. The change of solution color (a). UV-vis absorption spectra (b) and Mass absorption efficiency (MAE) (c) of reaction solution collected at reaction time intervals of 1 h. The change of EEM fluorescence spectra (d). Time profiles of BT degradation, inorganic products formation ($SO_4^{2-}$, $NO_3^-$, and $Cl^-$), particles formation, and total organic carbon (TOC) concentration (e). Size distribution of nanoparticles formed at reaction time intervals of 1 h (f). The change of NTA images (g).**

3. Page 3, Line 70: There supposed to use ", and" before "vanillic acid".

**Response:** Taking the reviewer's advice, ", and" has been added before "vanillic acid"

in the revised manuscript.

4. Page 3, Line 88: There supposed to use "and" before "2-chlorobenzothiazole".

**Response:** Taking the reviewer's advice, we have added "and" before "2-chlorobenzothiazole" in the revised manuscript.

5. Page 3, Lines 85-90: The units of sodium hydroxide and perchloric acid solutions are supposed to be mol L$^{-1}$. The format of units needs to be used consistently.

**Response:** Taking the reviewer's advice, we have modified the unit "mol L$^{-1}$" for continuity in the revised manuscript.

6. Page 11, Line 266: The criteria used for molecular formula assignment from Orbitrap MS data should be explicitly described in this paragraph.

**Response:** We thank the reviewer for this suggestion. The explicit description of the criteria used for molecular formula assignment from Orbitrap MS data is indeed necessary. Following the reviewer's advice, we have added a sentence in the second paragraph of the revised manuscript. "The number of atoms ($^{12}$C ≤ 30, $^{1}$H ≤ 30, $^{16}$O ≤ 20, $^{14}$N ≤ 10, $^{32}$S ≤ 10 and $^{35}$Cl ≤ 5) and a constraint of 4 ppm mass tolerance are applied as constraints to obtain the elemental composition."

7. Sec. 3.3 and Sec. 3.4: Although the presence of oligomers and nanoparticles is well established, quantitative yield data (e.g., mass-based or molar-based) would strengthen the case for atmospheric relevance. If not available, this limitation should be acknowledged.

**Response:** We appreciate the reviewer's comment on the importance of quantitative yield data to strengthen the atmospheric relevance of our findings. The current study is limited in quantifying mass- or molar-based yields of oligomers mainly due to the lack of standard substances. We did not perform LC-Orbitrap MS to complete quantitative analysis under the present experimental condition. The comparison of identified peaks of oligomers from Orbitrap MS with the formation of nanoparticles is well worth studying and this will be considered in our further experiments. We have explicitly

acknowledged the limitation of quantification in the revised manuscript, which has been added in the third paragraph of Section 3.4. "It should be noted that in this study the LC-Orbitrap MS analysis provides more qualitative insights, as quantitative determination was limited and not conducted. It can be considered and developed for further characterization of the formation and evolution of organic products."

8. The Introduction establishes that BTs themselves are potentially harmful to humans but does not foreshadow any health implications of the secondary products formed by BTs in the atmosphere. There is a mild logical disconnect between the Introduction and Conclusion on the topic of health implications. The Conclusion goes beyond what the Introduction prepares the reader for: it argues that aqueous-phase oxidation of BTs can impact climate and human health. This is a significant point – it implies the secondary aerosol products of BTs contribute to fine particulate pollution that humans could inhale, thereby affecting health. To ensure logical flow, the Introduction should at least hint that studying BTs' atmospheric oxidation is relevant not just for aerosol science but also for evaluating potential health impacts. The Introduction could include one or two sentences foreshadowing the potential health implications of BTs' oxidation products. Revise this.

**Response:** We thank the reviewer for this insightful suggestion. We have revised the Introduction to briefly hint at the health relevance of BTs' oxidation products. The last sentence at the third paragraph of the Introduction was modified to hint that studying BTs' atmospheric oxidation is relevant to evaluating potential health impacts. "However, the detailed aqueous-phase oxidation mechanisms of BTs in the atmosphere remain largely unexplored, hindering a holistic understanding of their atmospheric chemistry and potential health risks." This modification ensures a logical flow, preparing the readers in the Introduction for the Conclusion's discussion of the impacts of BT-derived secondary aerosols on human health.

9. The authors make a compelling case for the atmospheric relevance of BT oxidation, but the manuscript would benefit from a more detailed discussion of future research directions. For instance, what are the potential impacts on human health, or how can

**Response:** We thank the reviewer for the suggestion regarding future research directions. To evaluate the potential impacts on human health, it is first necessary to identify and quantify the key secondary products of BT oxidation in the ambient atmosphere. Subsequently, toxicological assessments of these key products should be conducted. The current study is limited in quantifying secondary products mainly due to the lack of standard substances, which further limits the selection of key products and their toxicological assessments. Thus, we have added a new sentence at the end of the revised Conclusion section. "Future research should also focus on selection of key secondary products from BTs oxidation in the ambient atmosphere and their toxicological assessments to evaluate the potential impacts of BTs oxidation on human health." This sentence explicitly calls for future research focusing on the toxicological evaluation of BT-derived secondary products in the atmosphere.

10. Page 17, Line 380: There supposed to use "BTs concentrations are comparable" instead of "is".

**Response:** Taking the reviewer's advice, we have corrected this mistake.

---

## Author Comment (AC3)

**Author's Response to Referee #3**

We greatly appreciate the time and effort that Referee #3 has devoted to reviewing our manuscript. The comments are thoughtful and helpful in improving the quality of our paper. Below we make a point-by-point response to these comments. The response to Referee #3 is structured in the following sequence: (1) comments from the referee in blue color, (2) our response in black color, and (3) our changes in the revised manuscript in red color.

In this paper, Zhang et al. measure the reactivity in water of a specific class of aromatic organic compounds, namely benzothiazoles (BTs).

Since BTs are industrially produced, elucidating their multiphase fate in the atmosphere is important. They used the relative rate methodology to experimentally determine the reaction rates of BTs with OH. The rates they found are in line with the literature for BT, which gives trust in the new measurements for MBT and CBT They carried out product formation experiments (IC for inorganics, Orbitrap for organics, and NTA for nanoparticles) and measured optical properties (fluorescence spectrophotometry) to propose an OH-reaction mechanism of BTs. This last part is particularly interesting for understanding the contribution of organics aqueous phase chemistry to SOA formation for instance, and organics atmospheric aging in general.

Overall the paper is good and could be published in ACP after the authors address these comments as they deem appropriate.

Major comments

1. - Sect. 2.3: These types of experiments are often carried out in triplicates to account for systematic uncertainty. Maybe I missed the mention of it but if the experiments were not repeated, it would be important to explain why.

**Response:** We thank the reviewer for raising the point regarding the need for repeated kinetic experiments. We fully agree that performing the kinetic experiments in triplicate is essential to account for systematic uncertainties and to ensure reliable results. Our

experiments were in fact carried out in triplicate. Sect. 2.3 already contains a statement indicating the experiments were repeated. "Kinetic experiments were performed three times at initial pH 2 and 10 with the adjustment of the concentration ratio of BTs and the reference compound for accurate determination of the rate constants for the reaction of BTs with OH radicals." As shown in Table 2, the rate constants for the reactions of BTs with each reference compound were determined based on three replicate runs. The ratio of initial concentrations of BTs to reference compounds were adjusted with 1:1, 1:2, and 2:1 (Table S2 in the Supplement).

However, we recognize that this detail might not have been sufficiently clear to the reader. To avoid any misunderstanding, we have slightly revised the wording of that sentence in Section 2.3 to explicitly highlight the triplicate nature of kinetic experiments as follows: "Kinetic experiments were performed in triplicate at initial pH 2 and 10, with the concentration ratio of BTs to the reference compounds adjusted to ensure accurate determination of the rate constants for the reactions of BTs with OH radicals." This update makes it unambiguous that all kinetic experiments were repeated three times, thus directly addressing the reviewer's concern about systematic uncertainty.

2. - l. 191: Is there an explanation why previous reported rate constants are significantly lower?

**Response:** We thank the reviewer for this observation. The discrepancy between our measured rate constant ($\sim 9 \times 10^9$ $M^{-1}$ $s^{-1}$) and the lower values reported by Borowska et al. (2016) and Andreozzi et al. (2001) is indeed noteworthy. The differences likely due to varying experimental conditions, including the reaction temperatures, the reaction vessels, the type and the intensity of the light source, et al. The reaction temperature is maintained at 298 K in our study, but 293 K in the two previous studies (Borowska et al., 2016; Andreozzi et al., 2001). Lower temperature can generally lead to lower rate constant according to the Arrhenius equation. The reaction vessel used in our study is a quartz glass vessel, but ordinary glass vessels in the two previous study. Ordinary glass blocks more UV light than quartz glass. In addition, Borowska et al. (2016) employed a polychromatic mercury lamp while Andreozzi et al. (2001) used a

low-pressure 254 nm UV lamp. Differences in material of the reaction vessels and irradiation source can lead to different OH radical production rates. The methodology used to determine the rate constants also differed. Prior studies measured BT degradation indirectly or at higher initial BT concentrations, which can underestimate the true second-order rate constant. In Borowska et al. (2016), the initial BT concentration was on the order of $10^{-4}$ M, significantly higher than ~$10^{-6}$ M used in our study. Such high concentrations can cause increased competition for OH radicals, yielding lower measured rate constants. We added the following sentences in Section 3.1 of the revised manuscript for the explanation of the discrepancy, "The lower values from previous studies are potentially due to the differences in experimental conditions. The reaction temperatures in the previous studies were 293 K, lower than that in our study. In addition, the material of vessel employed in the previous studies is ordinary glass, but quartz glass in our study. The reduction in UV light caused by ordinary glass, together with the higher initial concentration of BT, limits the availability of OH radicals. These factors thus potentially yield lower measured rate constants."

3. - Conclusion: The lifetime evaluation is very useful and welcomed. However, the OH aqueous concentrations used here may be criticized. The literature referenced in Table 3 is quite old and does not reflect the high uncertainty surrounding the estimation of steady-state aqueous OH concentrations. The Arakaki et al. (2013) study did a nice job revisiting estimations of aqueous OH. For instance, OH is around $10^{-15}$ M in their maritime sample. The review of Bianco et al. (2020) also provides a more recent view on this question. I encourage the authors to revise their calculations and possibly their conclusions based on the range of reported aqueous OH concentrations.

**Response:** We thank the reviewer for their affirmation and suggestion for the lifetime evaluation. Initially, the evaluation of BTs lifetimes relied solely on a single reference and considered only the average steady-state aqueous OH concentrations. Taking the reviewer's advice, we have modified Table 3 along with its associated text and references. We have incorporated both the average steady-state aqueous OH concentrations and their large ranges reviewed and evaluated in Herrmann et al. (2010),

Arakaki et al. (2013), and Bianco et al. (2020) for the BTs lifetime evaluation. The modified Table 3 in the revised manuscript is as follows:

**Table 3. Atmospheric lifetimes of BT, MBT, and CBT initiated by OH radicals in various aqueous and gas phases.**

| | Ref. | | $[OH]^a$ | $\tau_{BT}$ in $h^b$ | | $\tau_{MBT}$ in $h^b$ | | $\tau_{CBT}$ in $h^b$ | |
|---|---|---|---|---|---|---|---|---|---|
| Urban cloud droplets | Herrmann et al. (2010) & Bianco et al. (2020) | Mean | $3.5 \times 10^{-15}$ | 9.9 | 8.2 | 10.4 | 8.1 | 10.4 | 8.4 |
| | | Max | $1.6 \times 10^{-14}$ | 2.2 | 1.8 | 2.3 | 1.8 | 2.3 | 1.8 |
| | | Min | $2.9 \times 10^{-16}$ | 119.7 | 98.7 | 126.0 | 97.7 | 126.0 | 101.9 |
| | Arakaki et al. (2013) | Max | $1.9 \times 10^{-12}$ | 0.02 | 0.02 | 0.02 | 0.01 | 0.02 | 0.02 |
| | | Min | $1.0 \times 10^{-14}$ | 3.5 | 2.9 | 3.7 | 2.8 | 3.7 | 3.0 |
| Remote cloud droplets | Herrmann et al. (2010) & Bianco et al. (2020) | Mean | $2.2 \times 10^{-14}$ | 1.6 | 1.3 | 1.7 | 1.3 | 1.7 | 1.3 |
| | | Max | $6.9 \times 10^{-14}$ | 0.5 | 0.4 | 0.5 | 0.4 | 0.5 | 0.4 |
| | | Min | $4.8 \times 10^{-15}$ | 7.2 | 6.0 | 7.6 | 5.9 | 7.6 | 6.2 |
| | Arakaki et al. (2013) | Max | $2.4 \times 10^{-12}$ | 0.01 | 0.01 | 0.02 | 0.01 | 0.02 | 0.01 |
| | | Min | $2.6 \times 10^{-14}$ | 1.3 | 1.1 | 1.4 | 1.1 | 1.4 | 1.1 |
| Maritime cloud droplets | Herrmann et al. (2010) & Bianco et al. (2020) | Mean | $2.0 \times 10^{-12}$ | 0.02 | 0.01 | 0.02 | 0.01 | 0.02 | 0.01 |
| | | Max | $5.3 \times 10^{-12}$ | 0.01 | 0.01 | 0.01 | 0.01 | 0.01 | 0.01 |
| | | Min | $3.8 \times 10^{-14}$ | 0.9 | 0.8 | 1.0 | 0.7 | 1.0 | 0.8 |
| | Arakaki et al. (2013) | Max | $2.0 \times 10^{-12}$ | 0.02 | 0.01 | 0.02 | 0.01 | 0.02 | 0.01 |
| | | Min | $1.8 \times 10^{-13}$ | 0.2 | 0.2 | 0.2 | 0.2 | 0.2 | 0.2 |
| Urban deliquescent particles | Herrmann et al. (2010) | Mean | $4.4 \times 10^{-13}$ | 0.1 | 0.1 | 0.1 | 0.1 | 0.1 | 0.1 |
| | | Max | $1.9 \times 10^{-12}$ | 0.02 | 0.02 | 0.02 | 0.01 | 0.02 | 0.02 |
| | | Min | $1.4 \times 10^{-16}$ | 248.0 | 204.5 | 261.1 | 202.5 | 261.1 | 211.1 |
| | Arakaki et al. (2013) | | $8.0 \times 10^{-13}$ | 0.04 | 0.04 | 0.05 | 0.04 | 0.05 | 0.04 |
| Remote deliquescent particles | Herrmann et al. (2010) | Mean | $3.0 \times 10^{-12}$ | 0.01 | 0.01 | 0.01 | 0.01 | 0.01 | 0.01 |
| | | Max | $8.0 \times 10^{-12}$ | 0.004 | 0.004 | 0.005 | 0.004 | 0.005 | 0.004 |
| | | Min | $5.5 \times 10^{-14}$ | 0.6 | 0.5 | 0.7 | 0.5 | 0.7 | 0.5 |
| | Arakaki et al. (2013) | | $3.6 \times 10^{-12}$ | 0.01 | 0.01 | 0.01 | 0.01 | 0.01 | 0.01 |
| Maritime deliquescent particles | Herrmann et al. (2010) | Mean | $1.0 \times 10^{-13}$ | 0.3 | 0.3 | 0.4 | 0.3 | 0.4 | 0.3 |
| | | Max | $3.3 \times 10^{-12}$ | 0.01 | 0.01 | 0.01 | 0.01 | 0.01 | 0.01 |
| | | Min | $4.6 \times 10^{-15}$ | 7.5 | 6.2 | 7.9 | 6.2 | 7.9 | 6.4 |
| | Arakaki et al. (2013) | | $4.0 \times 10^{-16}$ | 86.8 | 71.6 | 91.4 | 70.9 | 91.4 | 73.9 |
| Gas phase | Prinn et al. (2001) | Mean | $1.0 \times 10^{6}$ | $39.7^c$ | $132.3^d$ | $15.1^c$ | | $56.4^c$ | |

| | | | | | | |
|---|---|---|---|---|---|---|
| Finlayson-Pitts and Pitts Jr. (2000) | Mid-day | $1.0 \times 10^7$ | $4.0^c$ | $13.2^d$ | $1.5^c$ | $5.6^c$ |

[a]The units of OH concentrations are M and molecule cm$^{-3}$ in the aqueous and gas phases, respectively.

[revised manuscript text omitted]

Minor comments

1. - l. 56: BT gas-phase oxidation is introduced before the presentation of Fig. 1 which shows which molecule BT specifically is

**Response:** We thank the reviewer for this notification. The BT in Line 56 refers to benzothiazole. We have used "benzothiazole" to replace "BT" before the presentation

of Fig. 1 in the revised manuscript for clarification.

2. - l. 60: the sentence is unusually formulated, it would be clearer to write something like "... could contribute to secondary organic aerosol after their oxidation into C3-8 organic compounds which also produces sulfuric acid"

**Response:** Taking the reviewer's advice, we have rewritten the sentence in the revised manuscript for clarification, "Simulation experiments of the gas-phase benzothiazole photooxidation with OH radicals indicated that benzothiazole could contribute to secondary organic aerosols after their oxidation into $C_{3-8}$ organic compounds and could also lead to the production of sulfuric acid."

3. - l. 264: this sentence is confusing, please reword it because I can't understand its meaning.

**Response:** Taking the Reviewer's advice, we have rewritten this sentence at the end of the first paragraph of Section 3.4 to explicitly state our point about the potential contribution of BTs oxidation to sulfate aerosol formation by combining observations from both aqueous- and gas-phase oxidation. "Considering both aqueous- and gas-phase oxidation results, atmospheric BTs oxidation appears to be a potential source of sulfate aerosols, given the significant sulfate yields observed from the oxidation of sulfur-containing structures."

4. - Fig. 4a: the histograms may be better understood if they were transposed, i.e. number of carbon on the x-axis, number of assigned formulas on the y-axis. No obligation, it's only a matter of taste.

**Response:** We appreciate the reviewer's suggestion regarding the presentation style of Fig. 4a. While transposing the histograms (placing the number of carbon atoms on the x-axis and the number of assigned formulas on the y-axis) might indeed offer an alternative visualization, we have deliberately retained the current arrangement to maintain consistency and coherence between Fig. 4a and Fig. 4b. Specifically, the horizontal presentation of experiments from left to right in Fig. 4b naturally aligns with the existing structure of Fig. 4a, enhancing readability across the figure set. We

therefore respectfully choose to keep the original presentation.

5. - l. 292: the sentence is confusing, please reword it. I guess it is supposed to mean that the CHONS fraction for BT and MBT is comparable to the CHONS+CHONSCl fraction for CBT?

**Response:** We thank the reviewer for highlighting the confusion in this sentence. The reviewer's interpretation accurately reflects our intended meaning. To improve clarity, we have rewritten the sentence in the third paragraph of Section 3.4 in the revised manuscript as follows: "In the CBT experiment, CHONS compounds alone account for only 30-40% of the total identified molecular formulas. However, when CHONSCl compounds are included, the combined proportion of CHONS and CHONSCl compounds becomes comparable to the fraction of CHONS compounds identified in the BT and MBT experiments."

**Response:** We thank the reviewer for the suggestion to include additional references. These references have been added according to our response to the third major comment.

---

## Author Comment (AC4)

**Author's Response to Referee #4**

We greatly appreciate the time and effort that Referee #4 has devoted to reviewing our manuscript. The comments are thoughtful and helpful in improving the quality of our paper. Below we make a point-by-point response to these comments. The response to Referee #4 is structured in the following sequence: (1) comments from the referee in blue color, (2) our response in black color, and (3) our changes in the revised manuscript in red color.

This manuscript presents an investigation into the atmospheric transformations of benzothiazoles (BTs) via OH radical-initiated oxidation in the aqueous phase. BTs are increasingly recognized as emerging environmental contaminants due to their extensive use in industrial and consumer products, and their widespread presence in urban environment, especially in air. While their gas-phase reactions have been previously studied, the aqueous-phase chemistry of BTs remains largely unexplored, despite its relevance in cloud water, fog, and deliquescent aerosols. This study addresses this knowledge gap. The authors quantified the second-order rate constants for BTs + OH reactions, characterized the reaction products via LC-Orbitrap MS, and monitored the formations of nanoparticles and light-absorbing chromophores. Key findings include the identification of multifunctional oligomers and the observation of brown carbon-like optical features among the products. These results suggest that BTs could be a previously underappreciated source of both inorganic and organic secondary aerosols in polluted environments. The integration of kinetic, molecular, and particle-level data represents a notable advance in understanding the atmospheric fate of this class of compounds. The study's focus is within the scope of ACP and the paper is well written, and I recommend an overall minor revision with a few comments below.

Comments:

1. Both the Abstract and Conclusion highlight the formation of atmospheric brown carbon from BTs' aqueous reactions as a key finding. However, the Introduction does not mention brown carbon or aerosol optical properties at all. The concept does not

appear until the Results and Discussion section, leading to a minor narrative gap. To harmonize, the Introduction could include a brief mention that secondary aerosols can include light-absorbing organic matter (i.e., brown carbon) that affects climate.

**Response:** We thank the reviewer for the suggestion. Mentioning the concept of brown carbon/aerosol optical properties in the Introduction enhances logical consistency and highlights the importance of aqueous-phase reactions. The relevant sentence in the third paragraph of the Introduction has been revised to explicitly state that aqueous-phase processing can generate light-absorbing compounds in secondary aerosols, which may influence climate. "Based on laboratory, modeling, and field studies conducted over the past decades, there has been increasing evidence that aqueous-phase processing affects the formation of both organic and inorganic components of secondary aerosols and contributes to the production of light-absorbing matter, which has the potential to influence climate (Lv et al., 2025; Mei et al., 2025; Li et al., 2023b; Zhang et al., 2018b; Ervens, 2015; Herrmann et al., 2015; Laskin et al., 2015; McNeill, 2015; Zhang et al., 2020a; Meagher et al., 1990)."

2. The terms, "secondary aerosol" vs. "secondary organic aerosol", are used mostly consistently, though a slight variation in usage may cause confusion. The Introduction refers to secondary organic aerosols (SOA) when giving examples of compounds that undergo aqueous processing, whereas the Abstract and Conclusion use the more general term "secondary aerosols" (which include both organic and inorganic components). The authors' intention is to show BTs contribute to both organic and inorganic matters in aerosols. To avoid confusion, the terminology should be made consistent. The Introduction could mirror this phrasing when framing the knowledge gap. The Introduction might state that "BTs may contribute to secondary aerosol (both organic and inorganic) via aqueous-phase reactions" instead of focusing only on SOA. This would ensure the reader knows from the start that both organic and inorganic aspects are of interest.

**Response:** We thank the reviewer for the suggestion on terminology. Stating that BTs may contribute to both organic and inorganic components of secondary aerosols

through aqueous-phase reactions is valuable, as it emphasizes the significance of investigating their aqueous-phase processing. In the revised manuscript, the sentence in the third paragraph of the Introduction now explicitly states that BTs may contribute to both organic and inorganic components of secondary aerosols through aqueous-phase reactions, due to their nature as sulfur-containing organic compounds. "Therefore, understanding the photooxidation reactions of BTs in the aqueous phase is significant not only for the integrated determination the atmospheric fate of BTs, but also for assessing their potential contributions to both organic and inorganic components of secondary aerosols due to their nature as sulfur-containing organic compounds."

3. 3.2: The optical data are well-presented, but their implications could be tied more explicitly to the evolving definition of atmospheric brown carbon. In other words, the authors should more directly connect their observed optical changes to the characteristics of brown carbon reported in the literature.

**Response:** We thank the reviewer for this insightful suggestion. In response, we have clarified the link between our observed optical properties and the evolving understanding of atmospheric brown carbon. Specifically, we addressed both the mass absorption efficiency at 365 nm ($MAE_{365}$) and the excitation-emission matrix (EEM) fluorescence features, which are key indicators of brown carbon, and compared them to those found in brown carbon-related studies.

After 5 h of photooxidation, the reaction solutions exhibited visible yellow-brown coloration and distinct optical changes. $MAE_{365}$ values reached 0.76 (Exp. BT-pH2), 0.74 (Exp. BT-pH10), 0.34 (Exp. MBT-pH2), 0.50 (Exp. MBT-pH10), 0.85 (Exp. CBT-pH2), and 1.02 $m^2$ $g^{-1}$ (Exp. CBT-pH10). These values fall within or approach the range of $MAE_{365}$ values for water-soluble brown carbon generated from biomass/coal combustion (0.21-3.1 $m^2$ $g^{-1}$; Zhang et al., 2024) and are comparable to those formed in laboratory-generated secondary brown carbon via aqueous Maillard-like reactions from carbonyl compounds mixed with ammonium sulfate or amine (0.15-4.5 $m^2$ $g^{-1}$; Tang et al., 2022b). These comparisons suggest that aqueous-phase oxidation of BTs can contribute to atmospherically relevant light-absorbing organic compounds.

EEM fluorescence spectra revealed clear red shifts in both excitation and emission wavelengths after 5 h of photooxidation. These spectral changes indicate an increase in molecular conjugation and complexity, which are characteristic of brown carbon chromophores. Notably, our results of EEM are quite close to the optical features of LO-HULIS (less-oxygenated humic-like substances), a subclass of atmospheric brown carbon identified in ambient aerosol studies (Jiang et al., 2022).

These combined findings support the classification of BT-derived products as atmospherically relevant brown carbon, both in terms of their absorption characteristics and molecular-level fluorescence signatures.

Thus, the results and discussions of MAE in Section 3.2 have been expanded and revised as follow: "The UV-vis absorption spectra were converted into mass absorption efficiency (MAE) using Eq. (3). After 5 h of photooxidation, the mass absorption efficiency at 365 nm ($MAE_{365}$) values of the light-absorbing substances derived from BTs reached 0.76, 0.74, 0.34, 0.50, 0.85, and 1.02 $m^2$ $g^{-1}$ for Exps. BT-pH2, BT-pH10, MBT-pH2, MBT-pH10, CBT-pH2, and CBT-pH10, respectively (Fig. 3c and Fig. S4). Among them, the products from CBT exhibit the highest $MAE_{365}$, resulting from the higher electronegativity of the -Cl group in CBT. $MAE_{365}$ values of the selected BTs-driven products are comparable to those reported for laboratory-generated brown carbon formed via Maillard-like aqueous-phase reactions from carbonyl compounds mixed with ammonium sulfate or amine, where $MAE_{365}$ ranged from 0.15 to 4.50 $m^2$ $g^{-1}$ depending on precursor combinations (Tang et al., 2022b). They are also within the $MAE_{365}$ range observed in ambient brown carbon samples from biomass and coal combustion sources, which span from 0.21 to 3.10 $m^2$ $g^{-1}$ (Zhang et al., 2024). It suggests that aqueous-phase oxidation of BTs can form light-absorbing organic products with optical properties relevant to atmospheric brown carbon." The results and discussions of EEM fluorescence spectra have been expanded and the following sentences have been added at the end of Section 3.2. "Moreover, EEM fluorescence features of BTs-driven products are consistent with the LO-HULIS (less-oxygenated humic-like substances) group, a subclass of atmospheric brown carbon identified in

studies of ambient aerosols (Jiang et al., 2022). LO-HULIS are less-oxygenated and nitrogen-containing aromatic compounds commonly observed in biomass burning aerosols. Given the spectral similarity, the aqueous-phase OH oxidation products of BTs in our study can be reasonably classified as LO-HULIS-type chromophores, further indicating a potential for contributing to the light absorption of aerosols.

**References**

Jiang, F., Song, J., Bauer, J., Gao, L., Vallon, M., Gebhardt, R., Leisner, T., Norra, S., and Saathoff, H.: Chromophores and chemical composition of brown carbon characterized at an urban kerbside by excitation–emission spectroscopy and mass spectrometry, Atmos. Chem. Phys., 22, 14971-14986, 10.5194/acp-22-14971-2022, 2022.

Tang, S., Li, F., Lv, J., Liu, L., Wu, G., Wang, Y., Yu, W., Wang, Y., and Jiang, G.: Unexpected molecular diversity of brown carbon formed by Maillard-like reactions in aqueous aerosols, Chem. Sci., 13, 8401-8411, 10.1039/d2sc02857c, 2022b.

Zhang, L., Li, J., Li, Y., Liu, X., Luo, Z., Shen, G., and Tao, S.: Comparison of water-soluble and water-insoluble organic compositions attributing to different light absorption efficiency between residential coal and biomass burning emissions, Atmos. Chem. Phys., 24, 6323-6337, 10.5194/acp-24-6323-2024, 2024."

4. One goal of the study is to elucidate reaction mechanisms. In the Introduction it explicitly says that mechanisms would be proposed based on the products. The paper discusses these mechanisms in detail in the main text. However, the Conclusion section, while summarizing outcomes (products formed, etc.), does not explicitly summarize the mechanism that was proposed. It implies a mechanism, e.g., "suggesting radical–radical oligomerization is responsible for oligomers and brown carbon formation", but it doesn't plainly state something like "a mechanism involving X, Y, Z steps was deduced." To fully close the loop, the authors could dedicate a sentence in the Conclusion to the mechanism. Including a high-level description of the reaction pathway would remind readers that the study achieved its aim of unraveling the

mechanism, not just finding the end products. This would tie back to the Introduction's promise of proposing a mechanism.

**Response:** We thank the reviewer for the suggestion. We have added a clear statement of the proposed reaction mechanism to the third paragraph of the Conclusion to ensure that the study's mechanistic insights are explicitly highlighted. This addition closes the loop between the Introduction's stated aim (to propose a mechanism) and the Conclusion, assuring the readers that we have indeed deduced a plausible reaction mechanism involving key steps. It reinforces that our study not only identified products but also elucidated how those products generate. "In this study, the formation of oxidized organic products with both higher and lower molecular weights was confirmed by LC-Orbitrap MS analysis, and a higher molar yield of sulfate was quantified by IC measurement. Based on these results, aqueous-phase oxidation mechanisms of BTs were proposed, in which OH radicals are assumed to initiate attack on the benzothiazole ring, forming radical intermediates that are subjected to radical-radical coupling, fragmentation, and further oxidation. It suggests that aqueous-phase BTs oxidation has the capacity to contribute to secondary aerosols, including both inorganic and organic components in the atmosphere. The contribution of secondary aerosols generated from BTs is considered non-negligible and can contribute to the fine particulate matters in some regions where BTs concentrations are comparable to common aromatic compounds."

5. The Abstract provides the exact rate constant values, highlighting the kinetic data, but it does not explicitly interpret what those mean in atmospheric terms. The Conclusion, on the other hand, emphasizes the implications of those kinetics by converting them to lifetimes and comparing aqueous vs. gas-phase transformation rates. While this is not a critical inconsistency (abstracts often omit detailed interpretation due to space), it is a difference in emphasis. Readers of the abstract see numbers but might not realize that corresponds to a very short lifetime. Abstract should include a brief interpretive phrase to complement the raw rate numbers. Similarly, the Introduction might also hint at expected fast kinetics by mentioning OH's known

reactivity. Currently, the Introduction imply this generally, and the conclusion confirms this quantitatively; making sure the abstract also reflects this would complete the chain of logic across all three sections.

**Response:** We agree with the reviewer's suggestion. For more consistency and clarity, we have revised the Abstract to include a brief interpretation that the reported rate constants are translated into lifetimes, immediately after listing the second-order rate constants for BT, MBT, and CBT. "Lifetimes ranging from several minutes to several hours were estimated under mean OH concentrations in various atmospheric aqueous phases, which are significantly shorter than those estimated under mean OH concentrations in the gas phase." This addition makes it clear that such rate constants imply rapid removal of BTs in the aqueous phase.

The Introduction already contains a general statement that aqueous-phase OH reactions can significantly shorten the lifetime of pollutants (Herrmann, 2003). With this revision, the Abstract, Introduction, and Conclusion are now aligned. The Introduction foreshadows OH's high reactivity, the Abstract presents the kinetic data along with its lifetime implication, and the Conclusion provides the quantitative comparison of aqueous- versus gas-phase lifetimes. This cohesive treatment across sections should help readers understand the atmospheric relevance of the kinetic findings.

6. The formation of sulfate is one of the major findings of this study, highlighted in both the Abstract and Conclusion, yet the Introduction does not mention sulfate at all. The Introduction should explicitly mention the possibility of sulfate generation from BTs during aqueous-phase oxidation. Establish a clearer rationale for investigating sulfate production, ensuring logical continuity from the Introduction's theoretical framework to the empirical results and their implications in the Conclusion.

**Response:** We thank the reviewer for the suggestion regarding explicitly mentioning sulfate formation in the Introduction. Clearly stating sulfate production potential significantly enhances the coherence and clarity of the manuscript. In the Introduction, we have noted that gas-phase BT reactions can lead to sulfate formation. In addition,

when responding to the reviewer's second comment, we revised the description of the significance of studying BTs' aqueous-phase reactions in the Introduction as follows: "Therefore, understanding the photooxidation reactions of BTs in the aqueous phase is significant not only for the integrated determination of the atmospheric fate of BTs, but also for assessing their potential contributions to both organic and inorganic components of secondary aerosols due to their nature as sulfur-containing organic compounds." This sentence implicitly indicates that aqueous-phase oxidation of BTs can also generate sulfate.

7. L235-248: The authors state that nanoparticles formed after 5 hours of photooxidation, with sizes ranging from 50 to 400 nm and concentrations ~10^8 particles/mL. This is a striking observation, indicating significant particle formation from BT oxidation. However, this result is presented with minimal discussion. This section should be revised to include additional discussion that situates the observed nanoparticle formation within the broader literature.

**Response:** We thank the reviewer for highlighting the importance of this observation. The formation of nanoparticles (50-400 nm, $\sim 10^8$ particles/mL) after 5 h of aqueous-phase photooxidation of BTs is quite a notable finding. To date, many studies have observed the formation of oligomeric and polymeric products in atmospheric relevant aqueous-phase reactions using mass spectrometry and spectroscopy (Li et al., 2023a; Li et al., 2023b; Tang et al., 2022b). These studies consistently demonstrate the generation of compounds with higher molecular weight, often linked to brown carbon formation. However, direct detection of nanoparticles in such systems remain scarce in previous studies. Our findings suggest that the aqueous-phase oxidation of BTs facilitates formation of nanoparticles, potentially from aggregation of oligomeric species. This observation may represent an underexplored pathway for secondary aerosols formation in cloud or particle water. Thus, we have expanded the corresponding discussion in Sect. 3.3 of the revised manuscript. "While previous studies have reported the formation of oligomeric and polymeric products in atmospheric relevant aqueous-phase reactions (Li et al., 2023a; Li et al., 2023b; Tang

et al., 2022b), direct detection of newly formed nanoparticles remains rare. The observed nanoparticles may originate from the aggregation of oligomeric products from aqueous-phase BTs oxidation, highlighting a potentially important but underrecognized route of secondary aerosols formation from aqueous-phase chemistry.

**References**

Li, F., Tang, S., Lv, J., He, A., Wang, Y., Liu, S., Cao, H., Zhao, L., Wang, Y., and Jiang, G.: Molecular-Scale Investigation on the Formation of Brown Carbon Aerosol via Iron-Phenolic Compound Reactions in the Dark, Environ. Sci. Technol., 57, 11173-11184, 10.1021/acs.est.3c04263, 2023a.

Li, F., Zhou, S., Du, L., Zhao, J., Hang, J., and Wang, X.: Aqueous-phase chemistry of atmospheric phenolic compounds: A critical review of laboratory studies, Sci. Total Environ., 856, 158895, 10.1016/j.scitotenv.2022.158895, 2023b.

Tang, S., Li, F., Lv, J., Liu, L., Wu, G., Wang, Y., Yu, W., Wang, Y., and Jiang, G.: Unexpected molecular diversity of brown carbon formed by Maillard-like reactions in aqueous aerosols, Chem. Sci., 13, 8401-8411, 10.1039/d2sc02857c, 2022."

8. L200-235: UV-vis and EEM data show red shifts and increased MAE values. Though the red shift is described and linked to conjugation, there is limited discussion of (1) how these light-absorbing products compare to known brown carbon chromophores, and (2) any atmospheric implications of higher MAE (e.g., warming effects).

**Response:** We thank the reviewer for this valuable comment. As noted, this point closely overlaps with Comment 3, which also addressed the interpretation of UV-vis and EEM optical data in the context of brown carbon characteristics. In response, we have substantially revised Section 3.2 to clarify how both the observed red shifts in EEM fluorescence and the increased MAE values correspond to known brown carbon chromophores and their atmospheric implications. Specifically, we compared our $MAE_{365}$ values with those reported for ambient and laboratory-generated brown carbon and discussed how the red-shifted EEM features resemble LO-HULIS-type chromophores. The potential for light absorption and radiative forcing was also

discussed. These revisions are described in detail in our response to the third comment, and the corresponding modifications of the manuscript have been incorporated in Section 3.2.

---

## Author Comment (AC5)

**Author's Response to Referee #5**

We greatly appreciate the time and effort that Referee #5 has devoted to reviewing our manuscript. The comments are thoughtful and helpful in improving the quality of our paper. Below we make a point-by-point response to these comments. The response to Referee #5 is structured in the following sequence: (1) comments from the referee in blue color, (2) our response in black color, and (3) our changes in the revised manuscript in red color.

Zhang et al. investigated the aqueous-phase OH reaction of benzothiazoles. The reaction kinetics, transformation products, reaction mechanism, and optical properties were reported. The atmospheric fate of benzothiazoles is not well illustrated at this point, this study highlights that the aqueous-phase oxidation of BTs can contribute to the secondary aerosol mass in the atmosphere, and have the potential to change the optical properties of ambient particles. I recommend this paper to be published after some revisions.

1. The focus of this study is aqueous-phase reaction of benzothiazoles. The atmospheric significance of this process depends on the concentrations of benzothiazoles in atmospheric droplets. The authors mention that the highest concentration of benzothiazoles in PM2.5 is at the level of ng/m3. I wonder what is concentration in cloud water or aerosol liquid water? Not all benzothiazoles in PM2.5 will be present in aerosol liquid water, so the concentration in aqueous-phase is likely to be lower than ng/m3. If the concentration of benzothiazoles in aerosol liquid water is low, this will limit the potential for aqueous-phase reactions to change the optical properties of atmospheric particles. The authors need to further discuss this point in Implication section.

**Response:** We thank the reviewer for highlighting this critical issue regarding BTs concentrations in atmospheric droplets. Explicitly addressing BTs concentrations in cloud, fog, and aerosol liquid water is significant to better assess the atmospheric implications of their aqueous-phase oxidation. There are rarely reported data on BTs

concentrations in cloud, fog, or aerosol liquid water samples to date. The available concentration of BT in atmospheric waters has been referenced in the Introduction of our manuscript. "BT is also present in urban rain, with a concentration as high as 70 ng $L^{-1}$ (Ferrey et al., 2018)." Furthermore, Liao et al (2021), referred in our manuscript, has reported the BTs concentration in $PM_{2.5}$ samples from Guangzhou and Shanghai, where climate is humid. According to the sampling dates from this paper, we summarized the relative humidity data at that time from ERA5 (Hersbach et al., 2023) as shown in Table R1. ERA5 is a widely adopted global atmospheric reanalysis dataset that integrates a vast array of historical observations with advanced data assimilation techniques, which is developed by the European Centre for Medium-Range Weather Forecasts. For most sampling dates in Guangzhou and Shanghai, daily average relative humidity was above 70%, and in some cases, it can reach up to 93%. Under these humid conditions, atmospheric aerosol particles typically contain a certain amount of moisture and even exist in deliquescent states. To a certain extent, the measured BTs concentrations in $PM_{2.5}$ under such humid conditions can reflect BTs levels in aerosol liquid water. However, the explicit concentrations of BTs in aerosol liquid water remain unknown and require further investigation. This knowledge gap limits our current understanding of the atmospheric implications. Thus, we have revised the second last sentence of the Conclusion section to add this future research direction explicitly. "In the future, research should focus on the occurrence and distribution of BTs in atmospheric aqueous phases, with particular emphasis on integrating real-world BT concentrations and varying ambient conditions to advance the understanding of their atmospheric chemistry."

**Table R1**. Basic information of the sampling sites (Liao, et al., 2021).

| Sampling sites | No. | Sampling date | Temperature (℃) | Relative humidity (%)[a] |
|---|---|---|---|---|
| Guangzhou | G1 | 2018/12/15 | 15.7 | 73.1 |
| | G2 | 2018/12/05 | 22.9 | 78.1 |
| (Winter, n=8; | G3 | 2018/12/21 | 22.9 | 83.5 |

| | | | | |
|---|---|---|---|---|
| Summer, n=6) | G4 | 2019/01/13 | 18.2 | 78.8 |
| | G5 | 2019/01/26 | 17.2 | 48.3 |
| | G6 | 2019/01/03 | 12.1 | 78.5 |
| | G7 | 2018/12/03 | 21.7 | 81.5 |
| | G8 | 2018/11/25 | 18.1 | 77.1 |
| | G9 | 2018/06/01 | 29.3 | 81.3 |
| | G10 | 2018/06/06 | 25.8 | 92.33 |
| | G11 | 2018/06/09 | 25.9 | 81.9 |
| | G12 | 2018/06/20 | 30.2 | 86.0 |
| | G13 | 2018/06/23 | 26.0 | 90.1 |
| | G14 | 2018/07/17 | 30.0 | 77.8 |
| | S1 | 2018/12/13 | 4.10 | 67.6 |
| | S2 | 2018/12/14 | 6.20 | 75.1 |
| | S3 | 2018/12/15 | 9.10 | 80.9 |
| | S4 | 2018/12/16 | 7.20 | 79.7 |
| | S5 | 2018/12/26 | 8.70 | 88.1 |
| Shanghai | S6 | 2018/12/27 | 5.60 | 67.3 |
| (Winter, n=8; | S7 | 2018/12/28 | 1.20 | 65.0 |
| | S8 | 2018/12/31 | 2.90 | 81.1 |
| Summer, n=6) | S9 | 2018/06/06 | 24.5 | 79.8 |
| | S10 | 2018/06/08 | 23.6 | 88.1 |
| | S11 | 2018/06/11 | 23.8 | 77.0 |
| | S12 | 2018/06/15 | 24.9 | 74.8 |
| | S13 | 2018/06/20 | 22.0 | 89.7 |
| | S14 | 2018/06/23 | 21.8 | 82.5 |

[a]Relative humidity data are obtained from ERA5 (Hersbach et al., 2023).

**References**

Hersbach, H., Bell, B., Berrisford, P., Biavati, G., Horányi, A., Muñoz Sabater, J.,

Nicolas, J., Peubey, C., Radu, R., Rozum, I., Schepers, D., Simmons, A., Soci, C., Dee, D., Thépaut, J-N. (2023): ERA5 hourly data on single levels from 1940 to present. Copernicus Climate Change Service (C3S) Climate Data Store (CDS), DOI: 10.24381/cds.adbb2d47 (Accessed on 51-May-2025)

Liao, X., Zou, T., Chen, M., Song, Y., Yang, C., Qiu, B., Chen, Z.-F., Tsang, S. Y., Qi, Z., and Cai, Z.: Contamination profiles and health impact of benzothiazole and its derivatives in PM2.5 in typical Chinese cities, Sci. Total Environ., 755, 142617, 10.1016/j.scitotenv.2020.142617, 2021.

2. Experiments were conducted at two different pH conditions (pH 2 and pH 10). The authors need to provide the rationale for selecting these two pH to perform experiments. Are they relevant to atmospheric conditions?

**Response:** We thank the reviewer for raising this question about our rationale for selecting pH 2 and pH 10 as experimental conditions. Both pH levels chosen for our experiments (pH 2 and pH 10) are relevant to atmospheric aqueous-phase conditions. As noted in Section 2.3 of our manuscript, atmospheric liquid phases exhibit a wide pH range from highly acidic to slightly alkaline. Herrmann et al. (2015) have summarized the range of pH in various aqueous phases. Most atmospheric aqueous environments are typically acidic, ranging from strongly (pH -2.5) to slightly (pH < 7) acidic conditions. Therefore, pH 2 was selected as a representative scenario for acidic atmospheric aqueous conditions, well-supported by numerous previous aqueous-phase simulation studies (Li et al., 2023). Under certain conditions, like haze and marine aerosols, aqueous particles can reach mildly alkaline states, with reported pH values ranging from >7 to 10. This mild alkalinity (pH 9-10) has also been frequently used in laboratory simulations of atmospheric aqueous-phase reactions (Witkowski et al, 2022, Tang et al., 2020). Thus, we selected pH 10 to represent such weakly alkaline atmospheric conditions.

**Reference**

Herrmann, H., Schaefer, T., Tilgner, A., Styler, S. A., Weller, C., Teich, M., and Otto,

T.: Tropospheric aqueous-phase chemistry: kinetics, mechanisms, and its coupling to a changing gas phase, Chem. Rev., 115, 4259-4334, 10.1021/cr500447k, 2015.

Li, F., Zhou, S., Du, L., Zhao, J., Hang, J., and Wang, X.: Aqueous-phase chemistry of atmospheric phenolic compounds: A critical review of laboratory studies, Sci. Total Environ., 856, 158895, 10.1016/j.scitotenv.2022.158895, 2023.

Tang, S., Li, F., Tsona, N. T., Lu, C., Wang, X., and Du, L.: Aqueous-phase photooxidation of vanillic acid: A potential source of humic-like substances (HULIS), ACS Earth Space Chem., 4, 862-872, 10.1021/acsearthspacechem.0c00070, 2020.

Witkowski, B., Jain, P., and Gierczak, T.: Aqueous chemical bleaching of 4-nitrophenol brown carbon by hydroxyl radicals; products, mechanism, and light absorption, Atmos. Chem. Phys., 22, 5651-5663, 10.5194/acp-22-5651-2022, 2022.

3. Some reference compounds such as suberic acid and toluic acid are organic acids. Will the pH of solution be changed when adding these acids into the solution?

**Response:** In the experiments for determining rate constants, suberic acid and toluic acid were used as reference compounds. Their addition did not alter the measured pH of the solutions obtained by a pH meter (SevenCompact, Mettler Toledo). This observation can be explained by their low concentrations and pKa values. The concentration of these organic acids as reference compounds was only 1-2 nM. Taking suberic acid as an example, it is a dicarboxylic acid. Even assuming complete ionization, a 2 nM suberic acid solution would contribute only $4 \times 10^{-9}$ M $H^+$. In comparison, the $H^+$ concentration is $10^{-2}$ M at pH 2, and the $OH^-$ concentration is $10^{-4}$ M at pH 10. Thus, the effect of $4 \times 10^{-9}$ M $H^+$ on the solution's pH is negligible and undetectable by the pH meter. Furthermore, suberic acid (pKa1 = 4.53, pKa2 = 5.52 at 25°C) and toluic acid (pKa = 4.37 at 25°C) are weak acids and do not fully dissociate in water. The actual $H^+$ concentration from their ionization is much lower than $4 \times 10^{-9}$ M. Therefore, the addition of these acids does not change the pH of the solution.

4. L185, the authors state that the rate constants of selected BTs reacted with OH radicals under the highly acidic condition were slightly lower than those under the

weakly alkaline condition. Why the reaction rate constants at low pH are lower than those under high pH conditions? Please clarify.

**Response:** We appreciate the reviewer's comment regarding the pH dependence of the rate constants. The slightly lower rate constants observed under highly acidic conditions ($8.0 \pm 1.8 \times 10^9 \text{ M}^{-1} \text{ s}^{-1}$) compared to weakly alkaline conditions ($9.7 \pm 2.7 \times 10^9 \text{ M}^{-1} \text{ s}^{-1}$) are within experimental uncertainty, and the difference is not statistically significant. This is why we stated in the text that the values were "slightly lower" rather than definitively different. In addition, one of the focuses of this study was to demonstrate that BTs react rapidly with OH radicals under atmospherically relevant pH conditions, leading to relatively short atmospheric lifetimes. While subtle pH-dependent variations in rate constants may exist, these minor differences were not a central focus of our investigation, as they do not substantially affect our main conclusions about the atmospheric fate of BTs.

5. It is interesting that nanoparticles were formed after OH reactions. Did you observe different patterns for the formation of nanoparticles for different BTs. I would expect to see different patterns given that their structures are different and the product composition prolife in Figure 4 is also different.

**Response:** The formation of nanoparticles is indeed an interesting observation, which is proposed from the aggregation of oligomeric products (Fig. 4). However, based on the nanoparticle characterization data (Fig. 3f and Fig. S8), while the size distributions of nanoparticles formed from different BTs share some common features, such as close size ranges and number concentration magnitudes. Distinct patterns among them cannot be conclusively identified. This limitation primarily stems from the nanoparticle tracking analysis (NTA) method employed in this study. As noted in Section 3.3, NTA provides only an approximate estimation of nanoparticles within a size range of 10-2000 nm. Moreover, nanoparticles may undergo agglomeration, deagglomeration, or partial dissolution during sample transfer, which can significantly influence measured particle concentrations and size distributions. Thus, the reported size distributions may not precisely reflect the true state of the nanoparticles in the reaction solution. While

this study focuses on demonstrating the potential for nanoparticle formation during the aqueous-phase reactions of BTs, a detailed comparison of nanoparticle patterns across different BTs was not a primary objective. Nevertheless, the reviewer's insightful suggestion highlights an important direction for future work. Further investigations employing more advanced characterization techniques would be valuable to elucidate whether structural differences among BTs lead to distinct nanoparticle formation patterns.

6. L274, The authors state that more products can be identified using ESI+ mode because organic products are rich in basic functionalities and slightly less rich in acidic functionalities. I am not sure this is correct for all organics or this is because BTs contain N, and N-containing compounds are more easily to be detected with ESI+ mode.

**Response:** We thank the reviewer for the comment regarding the interpretation of the +/-ESI results. The statement attributing the higher number of identified products in +ESI mode to the presence of basic functionalities may not be sufficiently rigorous. The causality should be softened to reflect a more nuanced interpretation. As observed by Tang et al. (2022), similar results were reported in Maillard-like reactions in atmospheric aqueous phases, where more compounds were identified in +ESI mode. They attributed the experimental result to the fact that products formed by Maillard reactions are rich in basic functional groups (e.g., amino) and poor in acidic functional groups (e.g., carboxyl), and are thus readily ionized in the +ESI mode. Lin et al. (2012) also showed that compounds detected in +ESI mode tend to contain reduced nitrogen functionalities like amines and alkaloids, which are basic in nature, whereas compounds identified in -ESI mode are often enriched in acidic groups such as carboxylic acids and organosulfates. Therefore, basic functionalities, especially organic compounds containing reductive nitrogen, generally tend to be detected in +ESI mode. As shown in Fig. 4b, although CHON species are more abundant in +ESI mode, the combined fraction of CHON and CHOSN species is comparable between modes, suggesting that the presence of nitrogen does not fully explain the observed differences between two modes either. Thus, the higher number of detected products in +ESI mode may result

from the general tendency of basic functionalities (not limited to N-containing species) to ionize more efficiently in this mode. The corresponding sentences in the second paragraph of Section 3.4 have been rewritten in the revised manuscript. "Meanwhile, products contained more identified compounds in positive-ion (+ESI) mode than in negative-ion (-ESI) mode, which suggests a possible enrichment of basic functionalities in the product mixture, as such groups are generally more amenable to ionization in +ESI mode (Lin et al., 2012)."

References

Lin, P., Rincon, A. G., Kalberer, M., and Yu, J. Z.: Elemental composition of HULIS in the Pearl River Delta Region, China: Results inferred from positive and negative electrospray high resolution mass spectrometric data, Environ. Sci. Technol., 46, 7454-7462, 10.1021/es300285d, 2012.

Tang, S., Li, F., Lv, J., Liu, L., Wu, G., Wang, Y., Yu, W., Wang, Y., and Jiang, G.: Unexpected molecular diversity of brown carbon formed by Maillard-like reactions in aqueous aerosols, Chem. Sci., 13, 8401-8411, 10.1039/d2sc02857c, 2022.

7. The proposed mechanism for BT reactions is convincing. I recommend the authors to describe the electron transfer process in Figure 5 such as ring-opening process, so that readers can better understand the chemical mechanism.

**Response:** We thank the reviewer for this helpful suggestion. The addition of the electron transfer process improves the clarity of the proposed mechanism. In the revised manuscript, we have modified Fig. 5 to include the electron transfer process at key steps, such as the ring-opening process, so that readers can better understand the chemical mechanism.

**Figure 5. Proposed reaction mechanism and structures of partial products of BT oxidized by OH radicals in the aqueous phase. Red structures with formulas and molecular weight are the products assigned in Orbitrap mass spectra.**

8. When discussing the atmospheric lifetime of BTs, the authors need to consider the impact of reaction conditions. The current calculation is based on an assumption that the pH of cloud droplets and deliquescent particles are similar to those using in the current experiments (pH 2 and pH 10). The authors need to point out the uncertainty associated with this assumption.

**Response:** We thank the reviewer for raising this important point. Estimation of BTs atmospheric lifetimes is influenced by rate constants and the aqueous-phase OH

concentrations (Eq. (4)). First, we used pH 2 and pH 10 as representative conditions to explore the rate constants of BTs reacted with OH radical in the aqueous phase. Our experimental results indicate that the reaction rate constants under the two pH conditions did not differ significantly. Second, OH concentrations in cloud and particle water cannot be directly measured and are typically derived from model simulations. As noted in the third major comment from Referee #3 and supported by the studies of Herrmann et al. (2010), Arakaki et al. (2013), and Bianco et al. (2020), OH concentrations in different aqueous media can span 2-4 orders of magnitude, influenced by multiple factors, including the composition and reactivity of dissolved organic compounds, the efficiency of gas-water transfer, and the presence of metal ions catalyzing OH formation (Wolke et al., 2005; Herrmann et al., 2005; Herrmann et al., 2010). It leads to substantial uncertainty in lifetime calculations. We have revised Table 3 and the corresponding discussion to reflect this variability and have emphasized the impact of OH concentration uncertainty on lifetime estimates in the response to the third major comment from Referee #3. Besides, model studies show that pH influences aqueous-phase OH concentrations via dissociation equilibria, transition-metal catalysis, and radical reactivity (Wolke et al., 2005; Herrmann et al., 2005). Thus, pH may affect lifetime estimation through its effect on OH radical levels. Assuming the pH of atmospheric aqueous phases is comparable to the experimental conditions may introduce more uncertainty in aqueous-phase OH radical concentrations, and further in the estimation of atmospheric lifetimes. Thus, we have noted this point in the Conclusion section in the revised manuscript to acknowledge this limitation. "It should be noted that the pH conditions in this study may introduce additional uncertainty into the estimated atmospheric lifetimes of BTs, due to the pH-dependent variability in aqueous-phase OH radical concentrations (Wolke et al., 2005; Herrmann et al., 2005)."

**References**

Arakaki, T., Anastasio, C., Kuroki, Y., Nakajima, H., Okada, K., Kotani, Y., Handa, D., Azechi, S., Kimura, T., Tsuhako, A., and Miyagi, Y.: A General Scavenging Rate Constant for Reaction of Hydroxyl Radical with Organic Carbon in Atmospheric

Waters, Environ. Sci. Technol., 47, 8196-8203, 10.1021/es401927b, 2013.

Bianco, A., Passananti, M., Brigante, M., and Mailhot, G.: Photochemistry of the Cloud Aqueous Phase: A Review, Molecules, 25, 423, 10.3390/molecules25020423, 2020.

Herrmann, H., Tilgner, A., Barzaghi, P., Majdik, Z., Gligorovski, S., Poulain, L., and Monod, A.: Towards a more detailed description of tropospheric aqueous phase organic chemistry: CAPRAM 3.0, Atmos. Environ., 39, 4351-4363, 10.1016/j.atmosenv.2005.02.016, 2005.

Herrmann, H., Hoffmann, D., Schaefer, T., Brauer, P., and Tilgner, A.: Tropospheric aqueous-phase free-radical chemistry: Radical sources, spectra, reaction kinetics and prediction tools, Chemphyschem, 11, 3796-3822, 10.1002/cphc.201000533, 2010.

Wolke, R., Sehili, A. M., Simmel, M., Knoth, O., Tilgner, A., and Herrmann, H.: SPACCIM: A parcel model with detailed microphysics and complex multiphase chemistry, Atmos. Environ., 39, 4375-4388, 10.1016/j.atmosenv.2005.02.038, 2005.